


# Increased vulnerability of European ecosystems to two compound dry and hot summers in 2018 and 2019

Ana Bastos[1], René Orth[1], Markus Reichstein[1], Philippe Ciais[2], Nicolas Viovy[2], Sönke Zaehle[1], Peter Anthoni[3], Almut Arneth[3], Pierre Gentine[4,5], Emilie Joetzjer[6], Sebastian Lienert[7], Tammas Loughran[8], Patrick C. McGuire[9], Sungmin O[1], Julia Pongratz[7], and Stephen Sitch[10]

[1]Max-Planck Institute for Biogeochemistry, Hans-Knöll Str. 10, 07745, Jena, Germany
[2]Laboratoire des Sciences du Climat et de l'Environnement, Gif-sur-Yvette, France
[3]KIT, Atmospheric Environmental Research, Garmisch-Partenkirchen, Germany
[4]Dept. of Earth and Environmental Engineering, Columbia University, NY 10027, USA
[5]Earth Institute and Data Science Institute, Columbia University, NY 10027, USA
[6]CNRM, Université de Toulouse, Météo-France, CNRS, Toulouse, France
[7]Climate and Environmental Physics, Physics Institute and Oeschger Centre for Climate Change Research, University of Bern, Bern, Switzerland
[8]Ludwig-Maximilian University, Geography Dept., Luisenstr. 37, 80333, Munich, Germany
[9]Department of Meteorology, Department of Geography Environmental Science, and National Centre for Atmospheric Science, University of Reading, Earley Gate, RG66BB Reading, UK
[10]College of Life and Environmental Sciences, University of Exeter, Exeter EX4 4RJ, UK

**Correspondence:** Ana Bastos (abastos@bgc-jena.mpg.de)

**Abstract.** In 2018 and 2019, central Europe was stricken by two consecutive extreme dry and hot summers (DH2018 and DH2019). The DH2018 had severe impacts on ecosystems and likely affected vegetation activity in the subsequent year, for example though depletion of carbon reserves or damage from drought. Such legacies from drought and heat stress can further increase vegetation susceptibility to additional hazards. Temporally compound extremes such as DH2018 and DH2019 can, therefore, result in an amplification of impacts by preconditioning effects of past disturbance legacies.

    Here, we evaluate how these two consecutive extreme summers impacted ecosystems in central Europe and how the vegetation responses to the first compound event (DH2018) modulated the impacts of the second (DH2019). To quantify the modulating role of vegetation responses to the impacts of each compound event, we first train a set of statistical models for the period 2001-2017 to predict the impacts of DH2018 and DH2019 on Enhanced Vegetation Index (EVI) anomalies from MODIS. These estimates can be seen as the expected EVI anomalies, had the impacts of DH2018 and DH2019 been consistent with past sensitivity to climate. These can then be used to identify modulating effects by vegetation activity and composition or other environmental factors such as elevated $CO_2$ or warming trends.

    We find two regions in which the impacts of the two DH events were significantly stronger than those expected based on previous climate–vegetation relationships. One region, largely dominated by grasslands and crops, showed much stronger impacts than expected in both DH events due to an amplification of their sensitivity to heat and drought, possibly linked to changing background $CO_2$ and temperature conditions. A second region, dominated by forests, showed browning from DH2018 to DH2019, even though dry and hot conditions were partly alleviated in 2019. This browning trajectory was mainly



explained by the preconditioning role of DH2018 to the observed response to DH2019 through legacy effects, and possibly by increased susceptibility to biotic disturbances, which are also promoted by warm conditions.

Dry and hot summers are expected to become more frequent in the coming decades posing a major threat to the stability of European forests. We show that state-of-the-art process based models miss these legacy effects. These gaps may result in an overestimation of the resilience and stability of temperate ecosystems in future model projections.

## 1    Introduction

Extreme dry and hot summers in western and central Europe have become more frequent over the past decades (Coumou and Rahmstorf, 2012; Seneviratne et al., 2014), a trend that is expected to continue as global mean temperatures rise (Barriopedro et al., 2011). Hot extremes in Europe are promoted by changes in atmospheric circulation (Coumou et al., 2015; Drouard et al., 2019) and amplified by strong feedbacks between the land-surface and the atmosphere, being therefore also associated with severe droughts (Miralles et al., 2014; Samaniego et al., 2018), i.e. compound hot and dry events (DH).

In Europe, DH events have usually strong negative impacts on ecosystems, such as reduced ecosystem productivity (Ciais et al., 2005; Bastos et al., 2020b). After severe drought and heat stress, recovery can be lagged, for example due to leaf shedding or non-reversible losses in hydraulic conductance (Ruehr et al., 2019). This, in turn may increase vulnerability if another DH occurs before complete recovery. A second DH event can additionally increase the hazard of other disturbances such as fires or insect outbreaks (Rouault et al., 2006; Pereira et al., 2005; Gouveia et al., 2016). More frequent extremes may, therefore,

threaten ecosystem stability by compounding multiple hazards and concurrent and lagged effects from highly impactful DH events.

At European scale, the summer in 2018 was the hottest since 1500 (Sousa et al., 2020) while at the same time leading to an unprecedented area affected by drought (Albergel et al., 2019; Bastos et al., 2020a). In 2019, central Europe was stricken by another extremely hot and dry summer (Boergens et al., 2020; Sousa et al., 2020). From a hydrometeorological perspective,

the dry and hot summers in 2018 and 2019 (DH2018 and DH2019, respectively) could be considered a temporally compound event (Zscheischler et al., 2020). For example, Boergens et al. (2020) have shown that while soil-moisture deficits in summer 2019 were not as pronounced as in 2018, total water storage was lower in 2019 due to the water storage deficit resulting from the 2018 event.

From an ecological perspective, these events are more complex as they constitute a combination of temporally (two consec-

utive extreme summers) and preconditioning (changes in ecosystem functioning) compound events. The DH2018 was one of the strongest in the past decades, leading to decreases in ecosystem productivity by up to 50% in central Europe (Bastos et al., 2020a; Buras et al., 2019). Compared to previous extreme summers, DH2018 was associated with increased sensitivity to temperature (Bastos et al., 2020b). This can be a sign of increased plant vulnerability to "hotter" droughts (Allen et al., 2015), or of





detrimental effects from increased growth in response to the previous sunny and warm spring (Bastos et al., 2020a). Given the

unprecedented magnitude of DH2018 and its severe impacts, it likely imposed such legacies throughout the subsequent year, for example through reduced growth or carbon reserve depletion that may have increased vulnerability to yet another dry and hot summer in 2019.

Repeated droughts have been linked to increased forest vulnerability in the northern mid-latitudes, although with variable responses (Anderegg et al., 2020). It remains unclear whether the increased vulnerability to a subsequent drought can be

explained by physical drivers (e.g. accumulated water-deficits or compound heat) or by modulating effects by vegetation responses during and following the first drought. Moreover, responses to drought are expected to be modulated by long-term increase in $CO_2$, but the direction of this effect is not clear: water-savings from reduced stomatal conductance should attenuate drought stress (Peters et al., 2018), but concurrent decrease in evapotranspiration cooling may amplify heat stress (Obermeier et al., 2018)

Separating the modulating effects controlled by vegetation responses to global change or by legacies from past disturbances (Kannenberg et al., 2020) and seasonal legacy effects (Buermann et al., 2018) in observations is problematic as it requires considering the compounding effects of multiple drivers (e.g., compound heatwave and drought) and separating the role of seasonal and inter-annual legacies both in physical variables (e.g., soil-moisture depletion) and in vegetation vulnerability to those drivers. Such effects have been separated for seasonal legacies using model experiments and counter-factual scenarios

((Lian et al., 2020; Bastos et al., 2020a)). Earth System models have been reported to fail at modelling woody biomass trajectories following droughts (Anderegg et al., 2015), but no simulations designed to isolate the individual impact of drought over subsequent years have been conducted. The simulations by land-surface models (LSMs) in Bastos et al. (2020a) separated the individual impact of DH2018 on carbon and water fluxes by using an additional factorial simulation. When extended to 2019, these simulations allow evaluating how models simulate inter-annual legacy effects of DH2018 and vulnerability to consecutive

droughts (DH2018 and DH2019).

The occurrence of two consecutive hot and dry summers is uncommon in central Europe but may become more likely in the coming decades Barriopedro et al. (2011); Boergens et al. (2020). Therefore, the DH2018 and DH2019 can provide insights on how resilient might European ecosystems be to repeated hot and dry summers in the coming decades. Here, we attempt to answer this question by using both observations and models to:

(i) evaluate the vulnerability of ecosystems to DH2018 and DH2019;

(ii) detect fingerprints of global change and disturbance legacy effects in ecosystem vulnerability to DH2018 and DH2019;

(iii) assess the ability of state-of-the-art LSMs to simulate the impacts on ecosystem of these two events.

Our results show an increasingly important contribution of vegetation condition, evaluated here by greenness and produc-

tivity, in preconditioning the response to both DH2018 and DH2019, suggesting that feedbacks between climate extremes and ecosystem functioning may increased vulnerability to climate change.





## 2 Data

### 2.1 Climate variables

For ecological studies, drought is better characterized by soil-moisture anomalies i.e. agricultural drought (Sherriff et al., 2011; Seneviratne et al., 2012; Samaniego et al., 2018) than atmospheric drought indices. We therefore base our drought assessment on two complementary soil-moisture datasets. The first is the observation-based soil moisture data obtained from SoMo.ml (O and Orth, 2020), used as reference in this study, and the second, for comparison with SoMo.ml, is give by ERA5 volumetric soil-water content (Hersbach et al., 2020).

The SoMo.ml data are generated using a Long Short-Term Memory neural network model trained with meteorological forcing from ERA5 and land surface characteristics as inputs and global in-situ soil moisture measurements (Dorigo et al., 2011; Zeri, 2020) as target variables. The data cover the period 2000-2019 and are available at 0.25°lat/lon resolution and daily time-steps. We remapped the fields to the finer resolution of the MODIS grid and aggregated the data to monthly means. We then subtracted the mean seasonal cycle and long-term linear trend and divided by the corresponding standard deviation to obtain standardized soil-moisture anomalies ($SM_{anom}$).

We use monthly temperature and volumetric soil-water content (layers 1 and 2, top 28cm) from the ECMWF ERA5 Reanalysis. ERA5 uses an improved land-surface data-assimilation system that makes use of remotely-sensed and *in-situ* observations, and shows improved skill compared to previous reanalyses (Hersbach et al., 2020) and good temporal agreement with a range of global soil-moisture networks (Li et al., 2020). Data were obtained from the Copernicus Climate Change Service at 0.25°lat/lon resolution (Hersbach et al., 2020) at monthly time-steps and selected for the period 2000-2019 (common with SoMo.ml) and remapped to the finer resolution of the MODIS grid using conservative regridding. Standardized anomalies were calculated by removing the mean seasonal cycle and long-term linear trend and then dividing by the corresponding standard deviation of temperature and soil-moisture fields ($T_{anom}$,$SM_{anom}^{ERA5}$). Soil-moisture anomalies from ERA5 are used for comparison of drought conditions with those estimated by SoMo.ml ($SM_{anom}$ for simplicity), although it is worth noting that the two datasets are not fully independent.

### 2.2 Vegetation and soil data

We used the 16-day Enhanced vegetation Index (EVI) from the Moderate Resolution Imaging Spectroradiometer (MODIS) sensor from the MOD13C1 CMG product. The MOD13C1 CMG follows a strict quality control and uses a gap-filling scheme to provide continuous cloud-free spatial composites from 1km data (Didan et al., 2015) projected on a 0.05°lat/lon grid. covering the period 2001–2019. MOD13C1 CMG is, therefore, a higher-quality product especially suitable for spatiotemporal analysis and for comparison with LSMs as intended here. EVI anomalies ($EVI_{anom}$) were calculated by removing the mean seasonal cycle and long-term linear trend, and were then scaled by the corresponding pixel-level temporal standard deviation. The standardization allows comparing the relative magnitude of anomalies for pixels with distinct temporal variability patterns and with vegetation productivity simulated by LSMs, which have different physical units.





For land-cover distribution we used the ESA Climate Change Initiative land-cover map (Kirches et al., 2014) (CCI-LC) for
the year 2018. The data are originally provided in land-cover classes at 300m spatial resolution and were converted to fractional
cover at 0.05°lat/lon resolution for forest, grassland, crop classes using the LC-CCI user–tool.

Isohydricity fields from global satellite measurements from Konings et al. (Konings et al., 2017) are available at 1°lat/lon res-
olution from https://github.com/agkonings/isohydricity. Anisohydric plants (low isohydricity) show weak regulation of stom-
atal opening, and prioritize carbon assimilation over water savings during droughts. High isohydric plants show strong stomatal
regulation of productivity and thereby preserve water at the cost of carbon assimilation during drought.

We use soil Available Water Capacity (AWC) from Ballabio et al. (2016) and Panagos et al. (2012), which used the Land
Use and Cover Area frame Statistical survey (LUCAS) topsoil database to map soil properties at continental scale. The data
are provided by the European Soil Data Centre (ESDAC) (esdac.jrc.ec.europa.eu).

### 2.3 Land-surface models

Standardized anomalies of gross primary productivity ($GPP_{anom}$) and soil-moisture ($SM_{anom}$) were estimated by the mean
of seven land-surface models (LSMs) between 1979–2019 from an extension of Bastos et al. (2020a) simulations: a baseline
simulation for comparison with observations and a factorial simulation to quantify the individual impact of summer 2018 and
its legacy effects, when compared to the reference simulation.

For the reference simulation, the models were forced with observed $CO_2$ concentration from NOAA/ESRL and changing
climate between 1979 and 2019 from ERA5 and fixed land-cover map from 2010 from LUH2v2 (Hurtt et al., 2011). An
additional simulation was ran where the models were forced with changing climate, except June–August 2018, where climato-
logical summer climate conditions were used to force the models as described in Bastos et al. (2020a). This simulation allows
evaluating the direct impact of DH2018 and its legacy effects. The model simulations were run for most models at 0.25 °spatial
resolution for the European domain (32–75°N and -11–65°E), following a spin-up to equilibrate carbon-pools. For more details
on the simulation protocol, we refer to (Bastos et al., 2020a). These simulations were extended here to 2019, using climate
fields from ERA5. Here we analyse gross primary productivity (GPP) and simulated soil-moisture.

The seven LSMs followed the protocol and extended the simulations in Bastos et al. (2020a) up to 2019. These models are:
ISBA-CTRIP (Joetzjer et al., 2015), JSBACH (Mauritsen et al., 2018), LPJ-GUESS (Smith et al., 2014), LPX-Bern (Lienert
and Joos, 2018), OCN (Zaehle et al., 2010), ORCHIDEE (Krinner et al., 2005) and SDGVM (Walker et al., 2017).

First, all models were remapped to a common 0.25 degree grid, and the multi-model ensemble mean was calculated for the
common period with MODIS (2001–2019). The variables were then deseasonalized, detrended and standardized as done for
the other variables in the study.



## 3 Methods

### 3.1 Drought characterization

We use the observation-based SoMo.ml as a reference dataset to define agricultural drought conditions. Regions with average $SM_{anom}$ below $-1\sigma$ (Seneviratne et al., 2012), during summer (JJA) are considered drought-affected areas during the DH events. Then, a regional domain affected by both DH2018 and DH2019 events is selected to evaluate the impacts of two consecutive DH events. Within this region most pixels had negative $SM_{anom}$ and the majority registered $SM_{anom} < -1.5\sigma$, but they differ in the magnitude of agricultural drought in 2019. This allows comparing responses across pixels for different

combinations of stress between DH2018 and DH2019.

### 3.2 Compound DH2018 and DH2019 events

#### 3.2.1 DH2018 and DH2019 impact characterization

To characterize different response "types" to DH2018 and DH2019, we group pixels in an unsupervised way (K-means clustering) based on the EVI impacts during the two extreme summers. Using an unsupervised method allows avoiding making

assumptions about the magnitude of impacts or the trajectory between DH2018 and DH2019 (DH2018→DH2019) when grouping pixels. For this, we applied a K-means cluster analysis (Hamerly and Elkan, 2003) using two features, corresponding to the $EVI_{anom}$ fields in DH2018 and DH2019, for pixels with negative $EVI_{anom}$ in DH2018. Four clusters captured the most significant differences in the impacts of DH2018 and corresponding DH2018→DH2019 responses: moderate/strong DH2018 impacts and moderate/strong impacts by DH2019. These clusters were then used to evaluate how LSMs simulate the summer

$GPP_{anom}$ and $SM_{anom}$.

#### 3.2.2 Detecting increased vulnerability to drought and heat stress

We propose that the two events can be considered a combination of temporally and preconditioning compound events (Fig. 1): a sequence of two DH events, whose impacts may be preconditioned by ecosystem vulnerability to DH, especially in the case of DH2019. The DH impacts and ecosystem vulnerability, i.e. the propensity to be negatively impacted by a given event, are

165 assessed by remotely-sensed EVI and modelled GPP anomalies.

The difference between the reference and factorial simulations by LSMs allow separating the modulating effects of DH2018 legacies to the DH2019 impacts (dashed arrow in Fig. 1). Separating the legacies in observations is more challenging, because the EVI signal at any time-step includes both signals. We hypothesise that preconditioning effects from legacies from past disturbance (modulating DH2019) but also from global change (modulating DH2018 and DH2019) should be detectable by

170 changes in ecosystem sensitivity to similar hazards. Increased vulnerability can thus be detected if $EVI_{anom}$ values are lower (more negative or less positive) than those expected for a given drought or temperature anomaly based on past sensitivities. Inversely, acclimation to drought could result in $EVI_{anom}$ being less negative or more positive than expected for a given $SM_{anom}$.


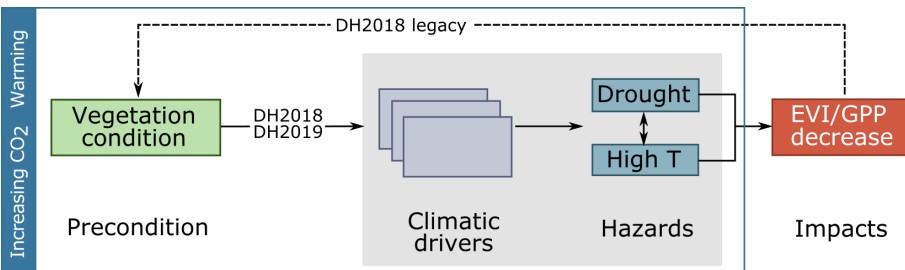

**Figure 1.** Conceptual description of the compound DH2018 and DH2019 events. Dry and hot conditions in both summers were a result of compouding atmospheric drivers (synoptic patterns, preceding climate anomalies, land-atmosphere interactions). The DH2018 impacts were modulated by seasonal legacy effects in ecosystem functioning from a sunny and warm spring. We hypothesise that legacies from the DH2018 event also contributed to modulate the response to DH2019. These impacts can be further modulated by long-term changes in ecosystem vulnerability to drought and heat stress due to anthropogenic climate change and increasing $CO_2$.

We test whether increased sensitivity to climate anomalies is detected for DH2018 and DH2019 in two steps: a linear case, and another including non-linear and seasonal legacy effects. In both steps, we characterize EVI-climate relationships for the period 2001–2017 and predict the EVI anomalies in DH2018 and DH2019. In the first step, we estimate the EVI-climate relationships for each cluster by univariate linear regression models of using growing-season $SM_{anom}$ or $T_{anom}$ as predictors. Because impacts on EVI could result from non-linear interactions between soil-moisture and temperature or from legacy effects from spring (Bastos et al., 2020a; Lian et al., 2020), we extend this analysis by random-forest (RF) regression using $SM_{anom}$ and $T_{anom}$ in spring (MAM) and summer $SM_{anom}$ or $T_{anom}$. We fit the RF model on a per-pixel basis but on 3x3 moving windows, in order to increase the sample size (i.e. $17 \times 9$) for each regression and reduce over-fitting. To further control for possible over-fitting and poor predictive skill outside of the training samples, we calculate the RF model out of bag scores. The importance of each predictor is then estimated by the Shapley additive explanation values (Lundberg and Lee, 2017).

The $EVI_{anom}$ predicted by the RF model for DH2018 and DH2019 correspond to the expected DH impacts if no changes ecosystem vulnerability to drought and heat were present, i.e. considering only links between atmospheric drivers, hazards and impacts in Fig. 1. It should be noted that the training period includes other DH events, particularly 2003 and 2015 (Ciais et al., 2005; Orth et al., 2016) thereby reducing the risk of attempting to predict anomalies out of the training sample. The difference between the RF model predictions and the actual $EVI_{anom}$ (model residuals) provides thus an estimate of the contribution of changes in ecosystem vulnerability to the DH2018 and DH2019 impacts.

For comparison with LSM simulations, the $EVI_{anom}$ clusters were remapped to 0.25 degree by largest area fraction calculation, and subsequently $GPP_{anom}$ and $SM_{anom}$ model ensemble means for each cluster were compared with corresponding $EVI_{anom}$ and ERA5 $SM_{anom}$. We first evaluate the linear relationships between the averaged $GPP_{anom}$ for each cluster and the corresponding climate anomalies. Then, we estimate the legacy effects from DH2018 to 2019 $GPP_{anom}$ based on the difference between the reference and factorial LSM simulations.


### 3.2.3  Modulating effects

To understand how land-cover can contribute to modulate the impacts of DH2018 and DH2019 we analyse the land-cover composition of each cluster. Given that central Europe is characterized mostly by mixed pixels, we do this by calculating the land-cover selectivity in each cluster for forests, natural grasslands and croplands. Selectivity is defined as the difference between the probability a given land-cover class being present within a cluster compared to its overall presence in the whole region. The probabilities are calculated by fitting a kernel-distribution function to the fractional cover fields for the whole region and for separate clusters. Positive (negative) selectivity means that a given land-cover type is more (less) common in a given cluster than its overall presence in the region.

Finally, we try to explain the changes in ecosystem vulnerability, which are given by departures of $EVI_{anom}$ residuals ($EVI_{anom}$ observed minus predicted) from the range of residuals in the training period. To do this, we evaluate how the spatial distribution of $EVI_{anom}$ residuals for DH2018 and DH2019 relates to climatic and ecological variables: $SM_{anom}$ and $T_{anom}$ in spring and summer, number of dry months in the year of the DH event and the preceding year (i.e. 2017–2018 for DH2018, and 2018–2019 for DH2019), $EVI_{anom}$ in the preceding summer $EVI_{anom}^{yr-1}$, forest, cropland and grassland cover fractions from CCI-LC, isohydricity (IsoH) and AWC. We include some of the drivers used to train the temporal climate-driven RF model to diagnose possible changes in the vulnerability to climate, i.e. the impact is still driven by climate conditions, but vegetation responds more strongly to climate anomalies than in the training period. $EVI_{anom}^{yr-1}$ is used to evaluate the preconditioning role of legacy effects from past extreme summers or disturbances (summer is the peak of the growing season in this region). The number of dry months and AWC (related to the maximum amount of water available for plants) are also included as they may explain non-linear relationships between $SM_{anom}$ and vegetation stress. Isohydricity provides a measure of the degree of stomatal regulation by plants. Since many of these variables have strong spatial covariability (e.g. $T_{anom}$ and $SM_{anom}$, or to some extent tree/grassland cover and IsoH), we evaluate their relationships with $EVI_{anom}$ residuals by calculating the partial rank correlation (Spearman's $\rho$) between each variable, controlling for the others.

To further evaluate long-term importance of inter-annual legacy effects in vegetation activity, we apply a second temporal RF model to $EVI_{anom}$ (Section 3.2.2) where we additionally include $EVI_{anom}^{yr-1}$ as a predictor of the regression. The model is trained for the period 2002–2017 also on 3×3 moving windows and spring and is then used to predict $EVI_{anom}$ in DH2018 and DH2019. The resulting model residuals were then compared to those of the climate–driven RF model.

## 4  Results

### 4.1  DH2018 and DH2019 impacts

Following the extreme summer in central Europe in 2018, mild temperatures and strong soil-moisture deficits remained until January 2019, when $SM_{anom}$ returned to normal conditions (Fig. A1, Fig. A2). In central Europe, June 2019 was extremely hot, but July and August 2019 were milder (Fig. A1, (Sousa et al., 2020)), and soil-moisture deficits were very pronounced in July (Fig. A2). In this region, excepting April 2019, the months preceding summer were not particularly dry and were



even slightly wetter than average in February, March and May, the latter also colder than average. Therefore, the DH2018 and DH2019 constitute more a sequence of two compound events than a long drought. The areas experiencing severe dry and hot conditions in both summers correspond to a region covering central and eastern Europe and southern Sweden. This region is 230 our study domain and indicated by the rectangle in Fig.2).

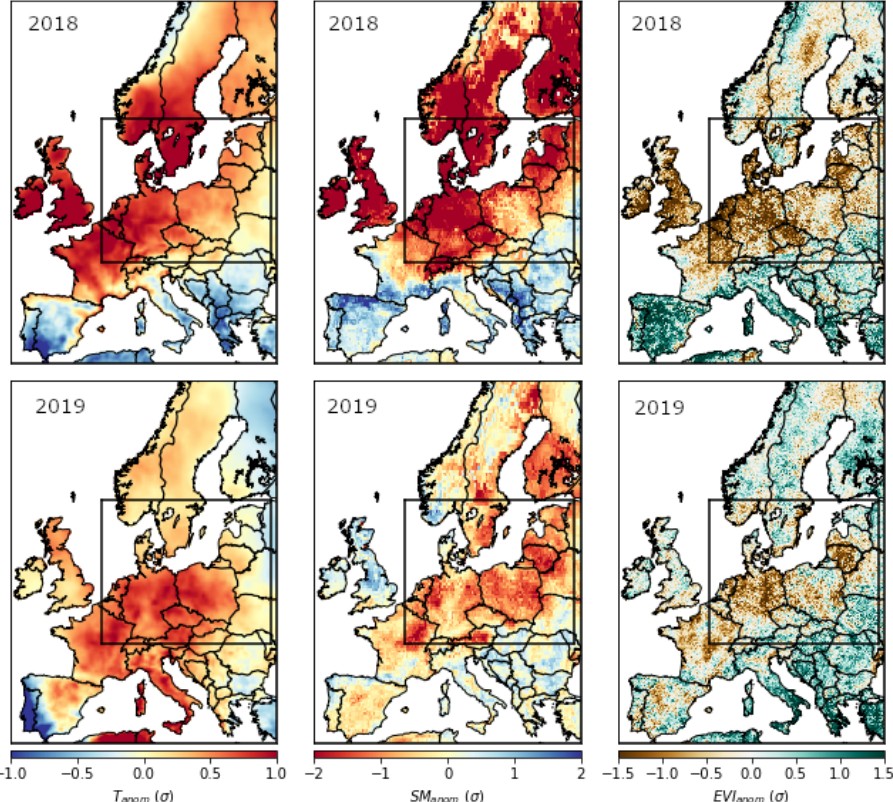

**Figure 2.** Spatial patterns of temperature ($T_{anom}$), soil-moisture ($SM_{anom}$) and EVI ($EVI_{anom}$) anomalies during summer 2018 (top panel) and summer 2019 (bottom panel) for the study region. The study region corresponds to a domain with dry and hot conditions in both 2018 and 2019 summers (DH2018 and DH2019).

Both DH events led to vegetation browning, though negative $EVI_{anom}$ were more widespread in DH2018 than DH2019. Within the study region, 79% of the area showing negative $EVI_{anom}$ in DH2018 ($EVI_{anom}^{DH2018}$) also registered negative $EVI_{anom}$ in DH2019 ($EVI_{anom}^{DH2019}$), although greening can be found in some areas. In this study, we limit our analysis to pixels negatively impacted by DH2018 and evaluate subsequent responses to DH2019 by grouping pixels based on 235 ($EVI_{anom}^{DH2018}$, $EVI_{anom}^{DH2019}$) values using unsupervised clustering. The spatial distribution of the resulting clusters is shown in Fig. 3 (left panel) and the corresponding ($EVI_{anom}^{DH2018}$, $EVI_{anom}^{DH2019}$) pairs are shown in the top right panel. For comparison, ($SM_{anom}^{DH2018}$, $SM_{anom}^{DH2019}$) and ($T_{anom}^{DH2018}$, $T_{anom}^{DH2019}$) pairs are also shown.





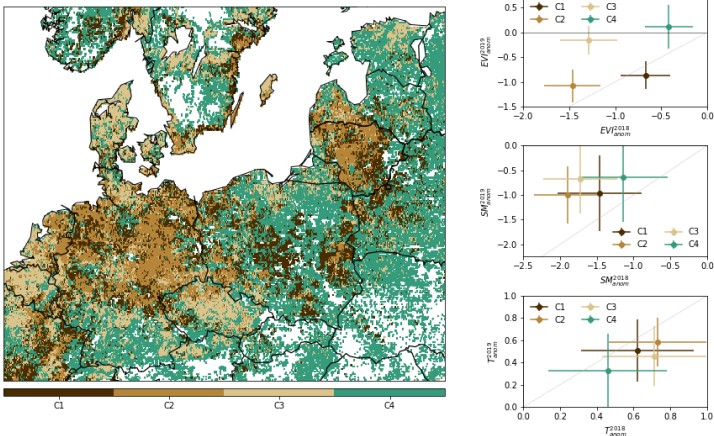

**Figure 3.** Classification of impact groups within the study region in central Europe. The left panel shows the spatial distribution of the four clusters from unsupervised classification of $(EVI_{anom}^{DH2018}, EVI_{anom}^{DH2019})$ values. The corresponding $(EVI_{anom}^{DH2018}, EVI_{anom}^{DH2019})$ distribution in each cluster are indicated in the top right panel (circles indicate the spatial mean and the lines spatial standard deviation within each cluster). The corresponding distribution of $SM_{anom}$ and $T_{anom}$ pairs are shown in the centre right and bottom right panels respectively. The grey line, indicates similar anomalies in the two DH events.

The clusters aggregate pixels according to different impacts in DH2018 and DH2019. Cluster C1 covers 20% of the area and includes pixels with moderate impacts in DH2018 and further browning in DH2019 ($EVI_{anom}^{DH2019}$ below the 1:1 line in Fig.3,

top right panel). This cluster is associated with mixed cover of forests (10-40%, dominated by needle-leaved) and grasslands (15-60%), (Fig.A3). Cluster C2 (15% of the area) corresponds to pixels experiencing strong impacts in both events and is associated with high grassland and cropland fractions and low forest cover. Pixels with strong impacts in DH2018 and partial recovery in DH2019 (C3, 21% of the area) are mainly dominated by croplands, while pixels showing positive $EVI_{anom}^{DH2019}$ (C4, 44%) correspond to mixed forest-grassland pixels (30-65% of forest, dominated by needle-leaved).

All clusters have negative $SM_{anom}$ and positive $T_{anom}$ in both DH events, but show alleviation of soil-moisture deficits and heat stress in DH2019 compared to DH2018 (Fig. 3). Clusters align along proportional $SM_{anom}$ and $T_{anom}$ in DH2018 vs DH2019 with overlapping distributions. The two recovery clusters (C3 and C4) correspond to pixels with less severe drought conditions and milder temperatures in DH2019, and C4 (greening) corresponds to pixels where dry and hot conditions in DH2018 were also more moderate. C2 corresponds to pixels experiencing drier and hotter anomalies in both summers and

shows accordingly stronger impacts. Cluster C1, however, shows increasing browning in DH2019 in spite of drought and heat stress alleviation (Fig.3), which suggests that the intensity of the hazard (temperature, drought) alone cannot account for the resulting impacts.





## 4.2 Ecosystem vulnerability to DH2018 and DH2019

We evaluate ecosystem vulnerability to the two compound events by comparing $EVI_{anom}$ in DH2018 and DH2019 with

past $EVI_{anom}$–$SM_{anom}$ and $EVI_{anom}$–$T_{anom}$ relationships (Fig. 4) for each cluster separately. All clusters show significant positive linear relationships between summer $EVI_{anom}$ and $SM_{anom}$ and negative linear relationships with $T_{anom}$ in 2001– 2017, consistent with a general summer water-limited regime. The relationships include the two extreme summers of 2003 and 2015 which had comparable $T_{anom}$ and $SM_{anom}$ to DH2018 and DH2019 in most clusters. The long-term sensitivities estimated are, though, robust even if these summers are excluded. All three clusters with strong impacts in one event (C1

DH2019, C3 DH2018) or both (C2) show $EVI_{anom}$ below the 95% confidence interval of the long-term linear relationship with both $SM_{anom}$ and $T_{anom}$ for that event. These departures are not likely explained simply by compounding heat and drought impacts, as $SM_{anom}$ and $T_{anom}$ in 2003 are similar, or even stronger, than in DH2018 and DH2019.

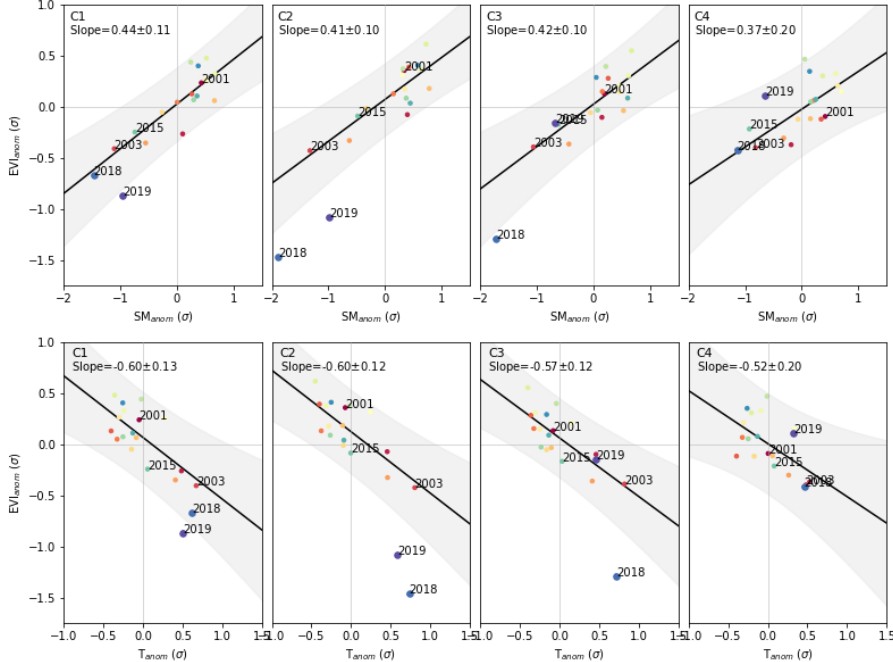

**Figure 4.** Departure of $EVI_{anom}$ in DH2018 and DH2019 from long-term climate-driven variability. Relationship between $EVI_{anom}$ and $SM_{anom}$ (top panel) and between $EVI_{anom}$ and $SM_{anom}$ (bottom panel) for each individual summer between 2001 and 2019 over the study region. The results are shown separately for the four clusters defined in Fig. 3. The black line and shaded areas show the relationship and respective 95% confidence intervals obtained by ordinary least-squares linear regression between $EVI_{anom}$ and the respective climate variable for all years between 2001–2017. Values of ($EVI_{anom}$, $SM_{anom}$) that deviate from the long-term relationships show increased sensitivity to climate anomalies, which can be a sign of increased vulnerability or degradation trajectories. The colours indicate individual years, ranging from 2001 (red) to 2019 (purple).



These departures may be related with seasonal legacy effects from the warm spring in DH2018 and could also be linked to non-linear responses to heat and drought under long-term changing environmental conditions. To account for these modulating effects, we model long-term (2001–2017) $EVI_{anom}$–climate relationships using spring and summer $SM_{anom}$ and $T_{anom}$ as predictors using random forest regression (see Section 3.2.2). For most pixels, the model is able to explain 48 –90% (median and maximum out of bag score) of the temporal variability of summer $EVI_{anom}$ in 2001–2017 (Fig.A4) and consistently estimates summer water limitation and negative legacy effects from spring warming (Bastos et al., 2020a; Lian et al., 2020).

In DH2018 and DH2019, the model has predominantly negative residuals, i.e. observed $EVI_{anom}$ is more negative or less positive than predicted from past vegetation–climate relationships, as found in the linear case (Fig. 5). Consistent with the results by the linear models, the residuals are below the range of the training period for the high impact clusters: C1 and C3 in DH2019 and DH2018, respectively, and C2 in both (Fig. 5, bottom panel). In the DH2019 "greening cluster" (C4), residuals are predominantly positive (i.e. observed $EVI_{anom}$ more positive than predicted), but still partly overlap with the range of residuals in the training period (Fig. 5).

We evaluate the role of diverse environmental variables in explaining the spatial distribution of residuals:

(i) the same climatic variables as used to train the RF model, indicating increased ecosystem sensitivity to climate;

(ii) $EVI_{anom}$ in the previous summer to account for inter-annual legacy effects;

(iii) isohydricity and land-cover composition to evaluate the modulating role of vegetation functioning differences;

(iv) soil available-water capacity and number of dry months, which can impose thresholds in water limitation.

To do this, we calculate the partial rank correlation of the spatial distribution of $EVI_{anom}$ residuals with respect to the explanatory variables selected (Fig. 6). Given the large number of pixels, all correlations are significant except those for croplands. In DH2018, vegetation condition in the previous year's growing season (i.e. summer 2017, +, positive effect), tree cover (+), $T_{anom}$ in spring ($T_{anom}^{spr}$, +) and summer $SM_{anom}$ ($SM_{anom}^{sm}$, -) show stronger relationships with $EVI_{anom}$ residuals. In DH2019, $EVI_{anom}^{yr-1}$ and $T_{anom}^{spr}$ are also the most relevant variables, but $EVI_{anom}^{yr-1}$ shows stronger correlation with $EVI_{anom}$ residuals than in DH2018, and $T_{anom}^{spr}$ has an opposite sign (-) In DH2019, $T_{anom}^{sm}$ and $SM_{anom}^{sm}$ show stronger relationships with $EVI_{anom}$ residuals as do the variables relating to water-availability (AWC, dry months and IsoH), all except $T_{anom}^{sm}$ with opposite sign as in DH2018.

To test whether the importance of $EVI_{anom}^{yr-1}$ is particular to the two DH events, or if it may reflect long-term inter-annual legacy effects of anomalies in vegetation activity, we fit a second temporal RF model where $EVI_{anom}^{yr-1}$ is used as an additional predictor (Figs. A4 and A6). Including vegetation condition in the previous summer improves the predictive power of the long-term RF model (72–97% out of bag score, compared to 48–90% for the model trained with climate drivers only). However, the residuals for DH2018 and DH2019 are comparable to those of the climate–driven model, while the residuals for the training period are considerably reduced.

## 4.3 DH2018 and DH2019 impacts simulated by LSMs

The GPP from the LSM multi-model ensemble mean matches well the relative differences in the impact of DH18 between clusters compared to $EVI_{anom}$ (Fig. 7, top and middle panels ). The temporal evolution of monthly GPP anomalies during the

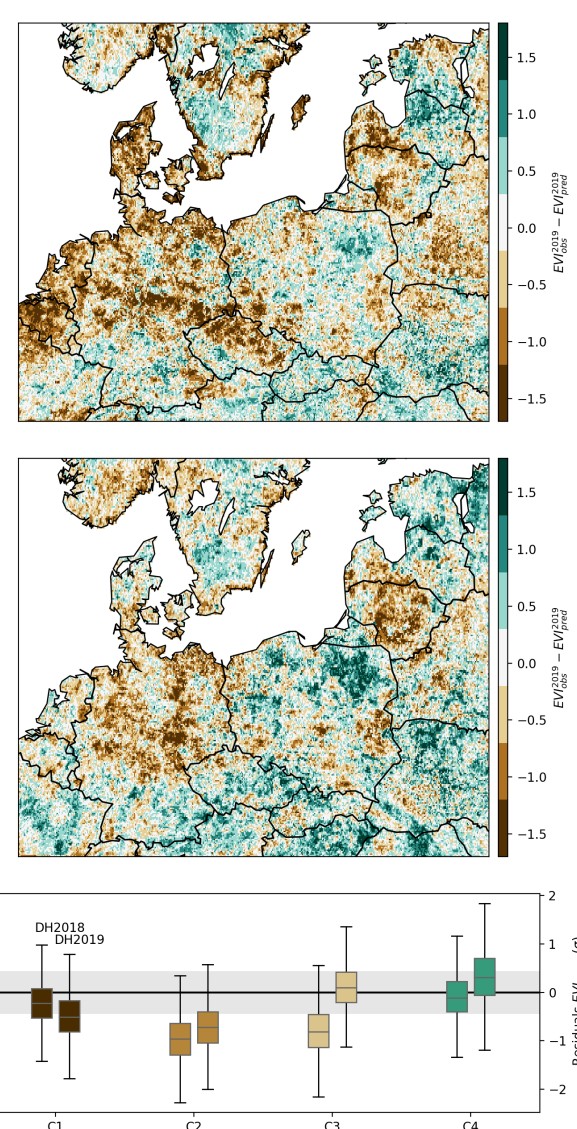

**Figure 5.** Spatial distribution of $EVI_{anom}$ residuals in DH2018 (top panel) and DH2019 (central panel) estimated by the temporal RF model trained for 2001–2017 with spring and summer $SM_{anom}$ and $T_{anom}$ as predictors. The corresponding distribution per cluster for each DH event is shown by the boxplots in the bottom panel. The shaded grey envelope indicates the range of residuals in the training period.





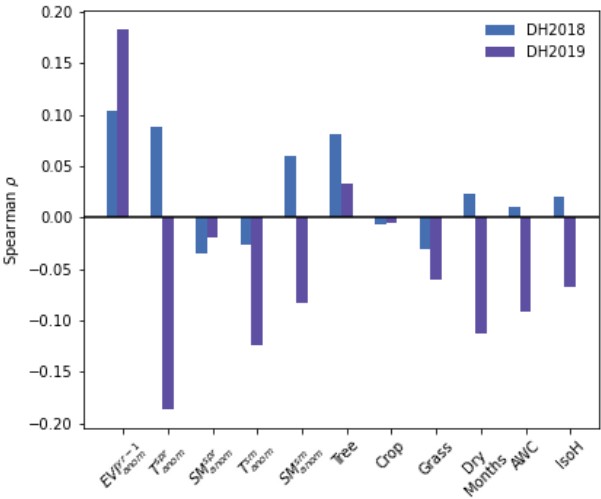

**Figure 6.** Spatial partial correlation (spearman) of $EVI_{anom}$ residuals with environmental variables in DH2018 and DH2019. The variables considered are: spring and summer $T_{anom}$ and $SM_{anom}$ (indicated by superscripts *spr* and *sm*, respectively), $EVI_{anom}$ in the previous growing season ($EVI_{yr-1}$), tree, crop and grassland cover, number of dry months, soil available water capacity (AWC) and plant isohydricity (IsoH). Because of the large number of pixels considered, all correlations are significant ($p-val << 0.01$), except for cropland cover.

2018 growing season (April to September, Table 1) also agrees with that of $EVI_{anom}$, with correlations between 0.74–0.90. However, 2019 trajectories from GPP simulated by LSMs indicate above-average spring and early summer productivity for all clusters, and strong positive $GPP_{anom}$ for both C3 and C4 during DH2019. However, correlations between $EVI_{anom}$ and 300 $GPP_{anom}$ are much lower in 2019 (-0.09 –0.43).

The disagreement in 2019 cannot be solely explained by errors in simulated soil-moisture anomalies ,since simulated $SM_{anom}$ shows very good agreement with both observation-based $SM_{anom}$ from SoMo.ml and $SM_{anom}^{ERA5}$ (correlations of 0.94–0.98, Table 1). LSMs simulate a stronger attenuation of drought compared to the observation-based $SM_{anom}$, though with consistent differences in $SM_{anom}$ between clusters (compare Fig. A7 and Fig. 3). The recovery simulated by LSMs in 305 2019 can be partly explained by too strong recovery of modelled soil-moisture (Fig. A7), but may also result from limited ability of LSMs in simulating changes in ecosystem vulnerability during the two DH events.

The sensitivity of $GPP_{anom}$ from LSMs to summer (JJA) soil-moisture anomalies (Fig. A8) is consistent with that of $EVI_{anom}$ in all clusters (Fig. 4). The sensitivity of $GPP_{anom}$ to temperature is also consistent with that of $EVI_{anom}$ for clusters C1 and C2, while for C3 and C4 LSMs estimate non-significant negative relationships between $GPP_{anom}$ and $T_{anom}$. 310 Simulated GPP agrees well with $EVI_{anom}$ during the 2018 growing season (r=0.74–0.90, Table 1, Fig. 7) and the deviation of $GPP_{anom}$ from the linear response for C2 and C3 in DH2018 are correctly captured by LSMs. In 2019, though, simulated growing season $GPP_{anom}$ shows low or even negative correlations with $EVI_{anom}$ in all clusters (r=-0.09–0.43). This cannot be explained by poor simulation of $SM_{anom}$, which shows high correlation to both observation-based soil-moisture datasets.





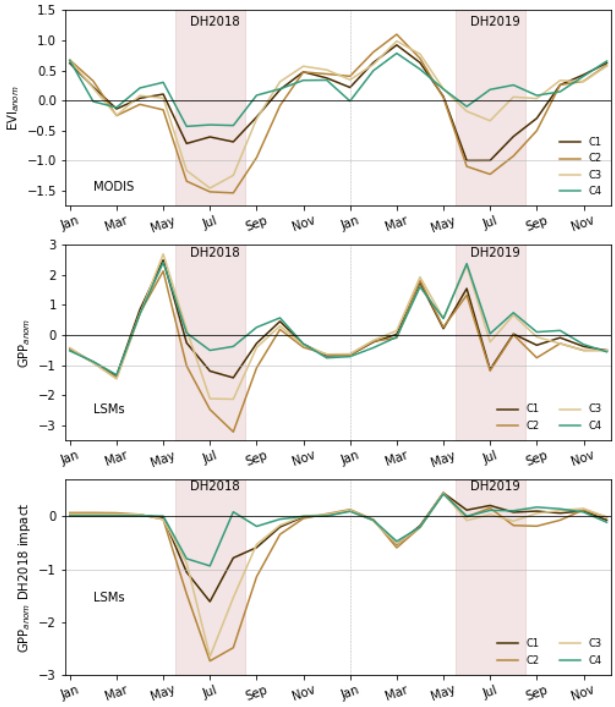

**Figure 7.** Observed and process-based model simulations of 2018/19 impacts. Seasonal evolution of $EVI_{anom}$ (top panel) and standardized GPP anomalies ($GPP_{anom}$, central panel) over the two year period for each cluster (defined in Fig. 3 and shown for LSM grid in Fig. A7). The bottom panel shows the difference between the reference and factorial simulations, and indicates the impacts of DH2018 on $GPP_{anom}$ simulated by models during the event and in the subsequent months until December 2019.

**Table 1.** Correlation of growing season (April–September) $SM_{anom}$ simulated by LSMs with $SM_{anom}$ from SoMo.ml and ERA5, and of $EVI_{anom}$ with GPP simulated by LSMs.

|  | C1 | C2 | C3 | C4 |
|---|---|---|---|---|
| $SM_{anom}$ 2018 | 0.98 | 0.98 | 0.97 | 0.97 |
| $SM_{anom}$ 2019 | 0.94 | 0.97 | 0.98 | 0.95 |
| $SM_{anom}^{ERA5}$ 2018 | 0.98 | 0.97 | 0.95 | 0.98 |
| $SM_{anom}^{ERA5}$ 2019 | 0.93 | 0.95 | 0.94 | 0.67 |
| $EVI_{anom}$ 2018 | 0.80 | 0.90 | 0.74 | 0.79 |
| $EVI_{anom}$ 2019 | 0.34 | 0.43 | 0.26 | -0.09 |





## 5 Discussion

### 5.1 Early signs of increased vulnerability

Our results indicate that the extremely negative $EVI_{anom}$ in response to DH2018 and DH19 cannot be predicted from past $EVI$–climate relationships, even when non-linear and seasonal legacy effects are considered (Figs. 4, 5). This suggests an important role of increased ecosystem vulnerability to climate (e.g., "hotter droughts") and of environmental factors (e.g. vegetation preconditioning effects), in explaining DH impacts.

The climatic variables used to train the temporal RF model still appear as relevant contributors to the spatial distribution of $EVI_{anom}$ residuals (Fig. 6) which supports a contribution of climate–driven increased vulnerability under the two extreme summers. The sign of the correlation between $T_{anom}^{spr}$ and $SM_{anom}^{spr}$ with $EVI_{anom}$ residuals indicates a positive effect of spring warming in partly offsetting vegetation growth (observed $EVIanom$ more positive or less negative than modelled), but with associated water depletion in spring amplifying the impacts of DH2018 in summer (Bastos et al., 2020a). This negative seasonal legacy effect through soil-moisture is also found in DH2019, but spring warming (or cooling, see Fig. A1) resulting in much stronger (weaker) impacts than those predicted by past vegetation–climate relationships (strong negative correlation). The positive correlation of $EVI_{anom}$ residuals in DH2018 with $SM_{anom}^{spr}$ and $T_{anom}^{spr}$ are also in line with increased sensitivity to water availability and temperature stress reported by Bastos et al. (2020b), i.e. stronger browning than predicted associated with warmer and drier conditions. In DH2019, the negative impact of summer $T_{anom}^{sm}$ was amplified, indicating increased vulnerability to heat stress, but the correlation of residuals with $SM_{anom}^{sm}$ is the opposite as in DH2018. This is consistent with the results for clusters C1 and C4, where browning in DH2019 happened in spite of drought alleviation, in the case of C1, or greening happened in spite of persisting soil-moisture deficits, in C4. However, these results indicate that other environmental effects may have an important role in modulating DH impacts, especially in DH2019.

The values of $EVI_{anom}$ in the previous summer show the strongest correlations with the residuals in both years (Fig. 6). The higher correlation between $EVI_{anom}$ residuals and $EVI_{anom}^{yr-1}$ in DH2019 than DH2018 points to a stronger contribution of legacy effects preconditioning the impacts of DH2019, resulting from the heat/drought stress imposed by DH2018. Even though considering inter-annual legacy effects mediated by vegetation condition improves the predictive skill of the RF model, this does not reduce the residuals in DH2018 and DH2019. The strong spatial relationship between $EVI_{anom}^{yr-1}$ with $EVI_{anom}$ residuals suggests that the preconditioning role of vegetation condition in DH2018 and DH2019 was amplified due a predom-inance of pixels with poorer vegetation condition before the DH events. This supports the important role of legacy effects from past stress conditions in impairing vegetation resistance to subsequent stressors, e.g. from to defoliation or damage from embolism (Ruehr et al., 2019) or higher susceptibility to diseases and pests due to reduced health (McDowell et al., 2020). The fact that $EVI_{anom}^{DH2018}$ is the most relevant predictor for residuals in DH2019 further supports the importance of damage from past heat/drought stress in amplifying the impacts of a subsequent DH event (Anderegg et al., 2020).

In both DH2018 and DH2019, higher tree cover fraction is associated with more positive or less negative residuals, indicating that trees buffered the impacts of DH conditions on ecosystem activity. This is consistent with the predominance of crops and grasslands in C2 and C3, which had strong negative residuals in DH2018, and of high tree cover in C4, where residuals are





mostly within the range of residuals in the training period and even slightly positive (Figs.4 and 5). Forests can better cope with drought with their deeper rooting depth (Fan et al., 2017) and through the use of carbon reserves to support activity

under stress conditions (Wiley, 2020). Moreover, some trees and grasses with stronger stomatal regulation can buffer the drought progression and its impacts by avoiding hydraulic failure (McDowell et al., 2020; Teuling et al., 2010). Even though isohydricity is strongly species-dependent (Konings et al., 2017), this effect is reflected in the small but positive effect of isohydricity in explaining DH2018 residuals.

The negative residuals in DH2019 for C2 are consistent with C2 showing the driest and hottest anomalies and predominantly

cropland cover, but in C1 the strongly negative $EVI_{anom}^{DH2019}$ are associated with mixed pixels (up to 40% forest and 20–60% grassland cover) and higher isohydricity. This points to increased vulnerability and degradation occurring mainly in natural ecosystems with stronger stomatal regulation, which is consistent with the negative relationship of isohydricity with $EVI_{anom}^{DH2019}$ residuals. The large negative $EVI_{anom}^{DH2019}$ residuals and browning in response to DH2019 in C1 may be linked to impaired growth due to damage from embolism, defoliation, or depletion of carbon reserves (Ruehr et al., 2019) under

longer drought conditions (negative effect of dry months). However, increased damage from heat stress due to reductions in evapotranspirative cooling (Obermeier et al., 2018) or cascading effects, such as the increased susceptibility of stressed trees to insect attacks and diseases cannot be excluded.

The stronger correlation of residuals with $EVI_{anom}^{yr-1}$ in DH2019, combined with favourable climatic conditions for insect growth (stronger negative effects of $T_{anom}^{sm}$ in DH2019, (Rouault et al., 2006)), indicates that such cascading effects could also

have contributed to amplify the impacts of DH2019. Results from field inventories and forest plots support this hypothesis. An increase in tree mortality and insect outbreaks in central Europe during 2018 has been reported (Schuldt et al., 2020). A recent assessment by the German Federal Minister for Food and Agriculture (BMEL, 2020) reported crown damage in 36% of all tree types in summer 2019, a 7% increase compared to 2018 and predominating in trees over 60 years of age. According to this report, the mortality rate in both needle-leaved and broad-leaved trees almost tripled from 2018 to 2019. Although no large

scale data on insect outbreaks is currently available, local authorities in regions where C1 is prevalent report increase in tree mortality from bark-beetle infestations: the Environment Ministry of North Rhine Westphalia in western Germany reported soaring rates of spruce affected by severe bark-beetle infestations, from about 1% in 2018 to over 12% in 2019 (MULNV-NRW, 2019). In the Czech Republic, rates of spruce damaged by bark-beetles more than tripled, leading to increased mortality (Hlásny et al., 2021). In Belgium, a "bark bettle task force" was created in September 2018 by the economic office of Wallonia

(OEW, 2018). Increased tree mortality and bark-beetle infestations have also been reported in eastern France (ONF, 2020).

## 5.2 Implications for earth system modelling

Temperate ecosystems are an important global sink of $CO_2$ (Pan et al., 2011) and are not usually considered hot-spots of drought risk and environmental degradation under climate change (Vicente-Serrano et al., 2020). Our results show that the past two extreme summers in central Europe reveal first signs of large-scale enhanced vulnerability in response to DH events (C2,

C3), and of potential degradation trajectories induced by consecutive events (C1). Even though limited to 20% of the study area, the patterns in C1 highlight the risks associated with more frequent and intense droughts and heatwaves expected in the





coming decades (Barriopedro et al., 2011; Boergens et al., 2020; Hari et al., 2020). At the same time progressive warming conditions are likely to promote compound occurrence of multiple disturbances, such as droughts and insect outbreaks, both promoted by warm and dry conditions. Interactions between compounding disturbances can further contribute to forest C losses
(Seidl et al., 2017; Kleinman et al., 2019). To anticipate such impacts, process-based modelling of ecosystem response to such events is needed.

The LSMs perform well in simulating productivity anomalies in 2018, but not in 2019. The comparison of the reference and factorial simulations allows showing that the poor performance in 2019 may related with interannual legacy effects. LSMs estimate legacies from DH2018 only in the early growing season (March to May 2019), but do not estimate any legacy effects
in summer (Fig. 7 bottom panel). The poor relationships between $EVI_{anom}$ and simulated $GPP_{anom}$ in response to DH2019 indicate that processes controlling legacy effects such as damage from embolism, carbon-starvation and resulting tree-mortality or disturbances induced by drought and heat such as insect outbreaks, currently missing in LSMs, likely explain the amplified impacts of DH2019.

LSMs are known to have limited ability to simulate drought-induced stress and tree mortality (Wang et al., 2012), and
lack impacts of biotic disturbances, although rudimentary approaches have been attempted (Kautz et al., 2018). These model shortcomings add to limitations in simulating soil-moisture variability and transitions between energy-limited and water-limited regimes. Attributing the LSM errors to specific climatic or non-climatic processes here is challenging since up-to-date datasets on tree mortality, tree carbon reserves or spatially-explicit information on biotic disturbances are very limited. Nevertheless, our results show that LSMs can simulate well the impacts of one strong drought (DH2018) on ecosystem dynamics but have
limited skill in simulating the impacts of a subsequent compound extreme event (DH2019) by missing important inter-annual legacy effects.

## 6   Conclusions

The two consecutive extreme dry and hot summers in central Europe (DH2018 and DH2019) had stronger impacts on vegetation activity than those expected by previous vegetation–climate sensitivity. This hints at large-scale increase in the vulnerability
of ecosystems to compound heat and drought events possibly modulated by vegetation responses to the long-term warming and increasing $CO_2$ trends. We find signs of degradation trajectories in 20% of the study area, where $EVI$ decreased even with drought alleviation in the following year. We attribute these trajectories to legacies from DH2018 amplifying the impacts of DH2019 which indicate that more frequent extreme summers may pose a major threat to the stability of temperate forests.

State-of-the-art land-surface models were able to simulate the exceptional impacts of DH2018, but they underestimated the
impacts of DH2019. This is explained by LSMs missing the preconditioning effect of DH2018 in DH2019 impacts as they cannot simulate inter-annual legacy effects from DH events on ecosystem activity. In addition, LSMs also lack representation of biotic disturbances, which are triggered by DH conditions and further promoted by plant stress in response to DH. Because DH events may become more common in the coming decades, overlooking these effects may result in an overestimation of the resilience of the $CO_2$ sink to climate change in temperate regions.



*Author contributions.* AB designed the study and methodology, conducted the data analysis and wrote the manuscript. RO, MR, PC, NV, and SZ contributed to initial development of study and to the first manuscript draft. SS and JP helped designing the LSM simulation protocol and SS coordinated the LSM modelling effort. SO provided the SoMo.ml dataset. PG contributed with expert knowledge. NV, SZ, PA, AA, PM, EJ, SL and TL ran the LSM simulations. All authors participated in the writing of the final version of the manuscript.

*Competing interests.* The autors have no competing interests

*Acknowledgements.* AB thanks Ulrich Weber for preprocessing of the MODIS data, Corinne Le Quéré for providing updated atmospheric CO2 fields for the model simulations. We thank the European Soil Data Centre (ESDAC), esdac.jrc.ec.europa.eu, European Commission, Joint Research Centre for making AWC data available and Alexandra Konings for providing the isohydricity dataset. AB received funding by the European Space Agency Climate Change Initiative ESA-CCI RECCAP2 project (ESRIN/ 4000123002/18/I-NB). SL has received funding from the European Union's Horizon 2020 research and innovation programme under grant agreement No 821003 (project 4C, Climate-Carbon Interactions in the Coming Century) and SNSF (grant no. $20020_1 72476$). RO and SO acknowledge support by the German Research Foundation (Emmy Noether grant 391059971). ORCHIDEE simulations was performed using HPC resources from GENCI-TGCC (grant 2020-A0070106328).





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

     Index and Phenology Lab, 2015.


Dorigo, W. A., Wagner, W., Hohensinn, R., Hahn, S., Paulik, C., Xaver, A., Gruber, A., Drusch, M., Mecklenburg, S., van Oevelen, P., Robock, A., and Jackson, T.: The International Soil Moisture Network: a data hosting facility for global in situ soil moisture measurements, Hydrol. Earth Syst. Sci., 15, 1675–1698, https://doi.org/10.5194/hess-15-1675-2011, 2011.

Drouard, M., Kornhuber, K., and Woollings, T.: Disentangling Dynamic Contributions to Summer 2018 Anomalous Weather Over Europe, Geophys. Res. Lett., 46, 12 537–12 546, https://doi.org/10.1029/2019gl084601, 2019.

Fan, Y., Miguez-Macho, G., Jobbágy, E. G., Jackson, R. B., and Otero-Casal, C.: Hydrologic regulation of plant rooting depth, Proceedings of the National Academy of Sciences, 114, 10 572–10 577, 2017.

Gouveia, C. M., Bistinas, I., Liberato, M. L., Bastos, A., Koutsias, N., and Trigo, R.: The outstanding synergy between drought, heatwaves and fuel on the 2007 Southern Greece exceptional fire season, Agricultural and Forest Meteorology, 218, 135–145, 2016.

Hamerly, G. and Elkan, C.: Learning the K in K-Means, in: Proceedings of the 16th International Conference on Neural Information Pro-
cessing Systems, NIPS'03, p. 281–288, MIT Press, Cambridge, MA, USA, 2003.

Hari, V., Rakovec, O., Markonis, Y., Hanel, M., and Kumar, R.: Increased future occurrences of the exceptional 2018-2019 Central European drought under global warming, Scientific Reports, 10, 12 207, https://doi.org/10.1038/s41598-020-68872-9, 2020.

Hersbach, H., Bell, B., Berrisford, P., Hirahara, S., Horányi, A., Muñoz-Sabater, J., Nicolas, J., Peubey, C., Radu, R., Schepers, D., Simmons, A., Soci, C., Abdalla, S., Abellan, X., Balsamo, G., Bechtold, P., Biavati, G., Bidlot, J., Bonavita, M., De Chiara, G., Dahlgren,
P., Dee, D., Diamantakis, M., Dragani, R., Flemming, J., Forbes, R., Fuentes, M., Geer, A., Haimberger, L., Healy, S., Hogan, R. J., Hólm, E., Janisková, M., Keeley, S., Laloyaux, P., Lopez, P., Lupu, C., Radnoti, G., de Rosnay, P., Rozum, I., Vamborg, F., Villaume, S., and Thépaut, J.-N.: The ERA5 global reanalysis, Quarterly Journal of the Royal Meteorological Society, 146, 1999–2049, https://doi.org/https://doi.org/10.1002/qj.3803, 2020.

Hlásny, T., Zimová, S., Merganičová, K., Štěpánek, P., Modlinger, R., and Turčáni, M.: Devastating outbreak of bark bee-
tles in the Czech Republic: Drivers, impacts, and management implications, Forest Ecology and Management, 490, 119 075, https://doi.org/10.1016/j.foreco.2021.119075, 2021.

Hurtt, G., Chini, L. P., Frolking, S., Betts, R., Feddema, J., Fischer, G., Fisk, J., Hibbard, K., Houghton, R., Janetos, A., et al.: Harmonization of land-use scenarios for the period 1500–2100: 600 years of global gridded annual land-use transitions, wood harvest, and resulting secondary lands, Climatic Change, 109, 117–161, 2011.

Joetzjer, E., Delire, C., Douville, H., Ciais, P., Decharme, B., Carrer, D., Verbeeck, H., De Weirdt, M., and Bonal, D.: Improving the ISBA (CC) land surface model simulation of water and carbon fluxes and stocks over the Amazon forest, Geoscientific Model Development, 8, 1709–1727, 2015.

Kannenberg, S. A., Schwalm, C. R., and Anderegg, W. R. L.: Ghosts of the past: how drought legacy effects shape forest functioning and carbon cycling, Ecology Letters, 23, 891–901, https://doi.org/10.1111/ele.13485, 2020.

Kautz, M., Anthoni, P., Meddens, A. J., Pugh, T. A., and Arneth, A.: Simulating the recent impacts of multiple biotic disturbances on forest carbon cycling across the United States, Global change biology, 24, 2079–2092, 2018.

Kirches, G., Brockmann, C., Boettcher, M., Peters, M., Bontemps, S., Lamarche, C., Schlerf, M., Santoro, M., and Defourny, P.: Land Cover CCI-Product User Guide-Version 2, ESA Public Document CCI-LC-PUG, 2014.

Kleinman, J. S., Goode, J. D., Fries, A. C., and Hart, J. L.: Ecological consequences of compound disturbances in forest ecosystems: a
systematic review, Ecosphere, 10, e02 962, https://doi.org/10.1002/ecs2.2962, 2019.

Konings, A., Williams, A., and Gentine, P.: Sensitivity of grassland productivity to aridity controlled by stomatal and xylem regulation, Nature Geoscience, 10, 284–288, 2017.





Krinner, G., Viovy, N., de Noblet-Ducoudré, N., Ogée, J., Polcher, J., Friedlingstein, P., Ciais, P., Sitch, S., and Prentice, I. C.: A dynamic global vegetation model for studies of the coupled atmosphere-biosphere system, Global Biogeochem. Cycles, 19, GB1015–, http://dx.
doi.org/10.1029/2003GB002199, 2005.

Li, M., Wu, P., and Ma, Z.: A comprehensive evaluation of soil moisture and soil temperature from third-generation atmospheric and land reanalysis data sets, Int J Climatol, 40, 5744–5766, https://doi.org/10.1002/joc.6549, 2020.

Lian, X., Piao, S., Li, L. Z., Li, Y., Huntingford, C., Ciais, P., Cescatti, A., Janssens, I. A., Peñuelas, J., Buermann, W., et al.: Summer soil drying exacerbated by earlier spring greening of northern vegetation, Science advances, 6, eaax0255, 2020.

Lienert, S. and Joos, F.: A Bayesian ensemble data assimilation to constrain model parameters and land-use carbon emissions, Biogeosciences, 15, 2909–2930, 2018.

Lundberg, S. M. and Lee, S.-I.: A unified approach to interpreting model predictions, in: Advances in neural information processing systems, pp. 4765–4774, 2017.

Mauritsen, T., Bader, J., Becker, T., Behrens, J., Bittner, M., Brokopf, R., Brovkin, V., Claussen, M., Crueger, T., Esch, M., et al.: Devel-
opments in the MPI-M Earth System Model version 1.2 (MPI-ESM 1.2) and its response to increasing CO2, Journal of Advances in Modeling Earth Systems, 2018.

McDowell, N. G., Allen, C. D., Anderson-Teixeira, K., Aukema, B. H., Bond-Lamberty, B., Chini, L., Clark, J. S., Dietze, M., Grossiord, C., Hanbury-Brown, A., Hurtt, G. C., Jackson, R. B., Johnson, D. J., Kueppers, L., Lichstein, J. W., Ogle, K., Poulter, B., Pugh, T. A. M., Seidl, R., Turner, M. G., Uriarte, M., Walker, A. P., and Xu, C.: Pervasive shifts in forest dynamics in a changing world, Science, 368,
https://doi.org/10.1126/science.aaz9463, 2020.

Miralles, D. G., Teuling, A. J., van Heerwaarden, C. C., and Vila-Guerau de Arellano, J.: Mega-heatwave temperatures due to combined soil desiccation and atmospheric heat accumulation, Nature Geosci, 7, 345–349, http://dx.doi.org/10.1038/ngeo2141, 2014.

MULNV-NRW: Waldzustandsbericht NRW 2019 (in German), techreport, Ministerium für Umwelt, Landwirtschaft, Natur- und Verbraucherschutz des Landes Nordrhein-Westfalen, 2019.

O, S. and Orth, R.: Global soil moisture from in-situ measurements using machine learning – SoMo.ml, preprint at https://arxiv.org/abs/2010.02374, 2020.

Obermeier, W. A., Lehnert, L. W., Ivanov, M. A., Luterbacher, J., and Bendix, J.: Reduced Summer Aboveground Productivity in Temperate C3 Grasslands Under Future Climate Regimes, Earth's Future, 6, 716–729, https://doi.org/10.1029/2018ef000833, 2018.

OEW: http://www.scolytes.be/, 2018.

ONF, O. N. d. F.: Epicéas, sapins, hêtres... Ces arbres qui souffrent de la sécheresse, Tech. rep., Office National des Fôrets, https://www.onf. fr/onf/secheresse-et-climat//4bd::ces-arbres-forestiers-qui-souffrent-de-la-secheresse.html, 2020.

Orth, R., Zscheischler, J., and Seneviratne, S. I.: Record dry summer in 2015 challenges precipitation projections in Central Europe, Scientific Reports, 6, 2016.

Pan, Y., Birdsey, R. A., Fang, J., Houghton, R., Kauppi, P. E., Kurz, W. A., Phillips, O. L., Shvidenko, A., Lewis, S. L., Canadell, J. G., Ciais,
P., Jackson, R. B., Pacala, S. W., McGuire, A. D., Piao, S., Rautiainen, A., Sitch, S., and Hayes, D.: A Large and Persistent Carbon Sink in the World's Forests, Science, 333, 988–993, https://doi.org/10.1126/science.1201609, 2011.

Panagos, P., Van Liedekerke, M., Jones, A., and Montanarella, L.: European Soil Data Centre: Response to European policy support and public data requirements, Land use policy, 29, 329–338, 2012.

Pereira, M. G., Trigo, R. M., da Camara, C. C., Pereira, J. M., and Leite, S. M.: Synoptic patterns associated with large summer forest fires
in Portugal, Agricultural and Forest Meteorology, 129, 11 – 25, https://doi.org/10.1016/j.agrformet.2004.12.007, 2005.



Peters, W., van der Velde, I. R., Van Schaik, E., Miller, J. B., Ciais, P., Duarte, H. F., van der Laan-Luijkx, I. T., van der Molen, M. K., Scholze, M., Schaefer, K., et al.: Increased water-use efficiency and reduced CO 2 uptake by plants during droughts at a continental scale, Nature geoscience, 11, 744, 2018.

Rouault, G., Candau, J.-N., Lieutier, F., Nageleisen, L.-M., Martin, J.-C., and Warzée, N.: Effects of drought and heat on forest insect
populations in relation to the 2003 drought in Western Europe, Annals of Forest Science, 63, 613–624, 2006.

Ruehr, N. K., Grote, R., Mayr, S., and Arneth, A.: Beyond the extreme: recovery of carbon and water relations in woody plants following heat and drought stress, Tree physiology, 39, 1285–1299, 2019.

Samaniego, L., Thober, S., Kumar, R., Wanders, N., Rakovec, O., Pan, M., Zink, M., Sheffield, J., Wood, E. F., and Marx, A.: Anthropogenic warming exacerbates European soil moisture droughts, Nature Climate Change, 8, 421–426, https://doi.org/10.1038/s41558-018-0138-5,
550    2018.

Schuldt, B., Buras, A., Arend, M., Vitasse, Y., Beierkuhnlein, C., Damm, A., Gharun, M., Grams, T. E. E., Hauck, M., Hajek, P., Hartmann, H., Hiltbrunner, E., Hoch, G., Holloway-Phillips, M., Körner, C., Larysch, E., Lübbe, T., Nelson, D. B., Rammig, A., Rigling, A., Rose, L., Ruehr, N. K., Schumann, K., Weiser, F., Werner, C., Wohlgemuth, T., Zang, C. S., and Kahmen, A.: A first assessment of the impact of the extreme 2018 summer drought on Central European forests, Basic and Applied Ecology, 45, 86–103, http://www.sciencedirect.com/
science/article/pii/S1439179120300414, 2020.

Seidl, R., Thom, D., Kautz, M., Martin-Benito, D., Peltoniemi, M., Vacchiano, G., Wild, J., Ascoli, D., Petr, M., and Honkaniemi, J.: Forest disturbances under climate change, Nature Climate Change, 7, 395–402, 2017.

Seneviratne, S. I., Nicholls, N., Easterling, D., Goodess, C., Kanae, S., Kossin, J., Luo, Y., Marengo, J., McInnes, K., and Rahimi, M.: Changes in climate extremes and their impacts on the natural physical environment, Managing the risks of extreme events and disasters to
advance climate change adaptation, pp. 109–230, 2012.

Seneviratne, S. I., Donat, M. G., Mueller, B., and Alexander, L. V.: No pause in the increase of hot temperature extremes, Nature Clim. Change, 4, 161–163, http://dx.doi.org/10.1038/nclimate2145, 2014.

Shepherd, T. G., Boyd, E., Calel, R. A., Chapman, S. C., Dessai, S., Dima-West, I. M., Fowler, H. J., James, R., Maraun, D., and Martius, O.: Storylines: an alternative approach to representing uncertainty in physical aspects of climate change, Climatic change, 151, 555–571,
565    2018.

Sherriff, R. L., Berg, E. E., and Miller, A. E.: Climate variability and spruce beetle (Dendroctonus rufipennis) outbreaks in south-central and southwest Alaska, Ecology, 92, 1459–1470, 2011.

Smith, B., Warlind, D., Arneth, A., Hickler, T., Leadley, P., Siltberg, J., and Zaehle, S.: Implications of incorporating N cycling and N limitations on primary production in an individual-based dynamic vegetation model, Biogeosciences, 11, 2027–2054, 2014.

Sousa, P. M., Barriopedro, D., García-Herrera, R., Ordóñez, C., Soares, P. M. M., and Trigo, R. M.: Distinct influences of large-scale circulation and regional feedbacks in two exceptional 2019 European heatwaves, Communications Earth & Environment, 1, 48, https://doi.org/10.1038/s43247-020-00048-9, 2020.

Teuling, A. J., Seneviratne, S. I., Stockli, R., Reichstein, M., Moors, E., Ciais, P., Luyssaert, S., van den Hurk, B., Ammann, C., Bernhofer, C., Dellwik, E., Gianelle, D., Gielen, B., Grunwald, T., Klumpp, K., Montagnani, L., Moureaux, C., Sottocornola, M., and Wohlfahrt, G.:
Contrasting response of European forest and grassland energy exchange to heatwaves, Nature Geosci, 3, 722–727, http://dx.doi.org/10.1038/ngeo950, 2010.

Vicente-Serrano, S. M., McVicar, T. R., Miralles, D. G., Yang, Y., and Tomas-Burguera, M.: Unraveling the influence of atmospheric evaporative demand on drought and its response to climate change, Wiley Interdisciplinary Reviews: Climate Change, 11, e632, 2020.



Walker, A. P., Quaife, T., van Bodegom, P. M., De Kauwe, M. G., Keenan, T. F., Joiner, J., Lomas, M. R., MacBean, N., Xu, C., Yang,
X., et al.: The impact of alternative trait-scaling hypotheses for the maximum photosynthetic carboxylation rate (Vcmax) on global gross
    primary production, New Phytologist, 215, 1370–1386, 2017.

Wang, W., Peng, C., Kneeshaw, D. D., Larocque, G. R., and Luo, Z.: Drought-induced tree mortality: ecological consequences, causes, and
    modeling, Environmental Reviews, 20, 109–121, https://doi.org/10.1139/a2012-004, 2012.

Wiley, E.: Do Carbon Reserves Increase Tree Survival during Stress and Following Disturbance?, Current Forestry Reports, pp. 1–12, 2020.

Zaehle, S., Friend, A. D., Friedlingstein, P., Dentener, F., Peylin, P., and Schulz, M.: Carbon and nitrogen cycle dynamics in the O-CN
    land surface model: 2. Role of the nitrogen cycle in the historical terrestrial carbon balance, Global Biogeochem. Cycles, 24, GB1006–,
    http://dx.doi.org/10.1029/2009GB003522, 2010.

Zeri, M.: A soil moisture dataset over the Brazilian semiarid region, https://doi.org/10.17632/XRK5RFCPVG.2, 2020.

Zscheischler, J., Martius, O., Westra, S., Bevacqua, E., Raymond, C., Horton, R. M., van den Hurk, B., AghaKouchak, A., Jézéquel, A.,
Mahecha, M. D., Maraun, D., Ramos, A. M., Ridder, N. N., Thiery, W., and Vignotto, E.: A typology of compound weather and climate
    events, Nature Reviews Earth Environment, 1, 333–347, https://doi.org/10.1038/s43017-020-0060-z, 2020.



# Appendix A: Supplementary Figures



**Figure A1.** Monthly temperature anomalies during 2018 and 2019. The rectangle indicates the study region.

**Figure A2.** Monthly soil-moisture anomalies during 2018 and 2019. The rectangle indicates the study region, i.e. the areas experiencing drought conditions ($SM_{anom} < -1\sigma$) during both DH2018 and DH2019. .



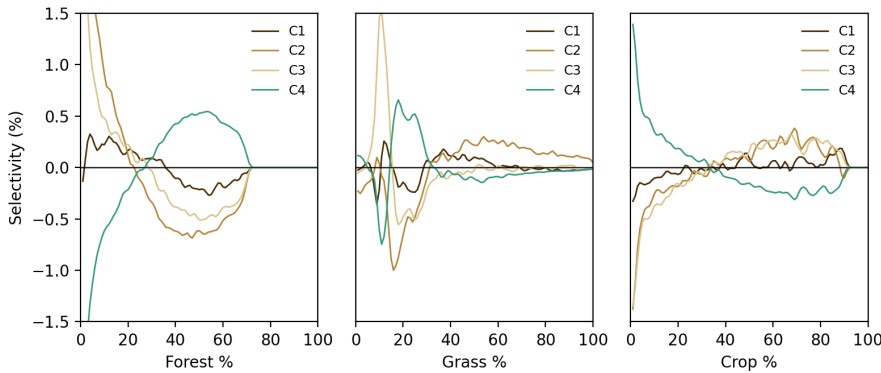

**Figure A3.** Selectivity of different land-cover composition for each cluster (Fig. 3). Selectivity is evaluated as the difference between the probability distribution of a given land-cover type (forest, left; grassland, middle; cropland, right) and the probability distribution of that land-cover type in the selected region. If selectivity is positive, the cluster is preferentially composed by the given land-cover type and the opposite for negative values. The 2018 land-cover classification maps from from ESA CCI-LC are used.



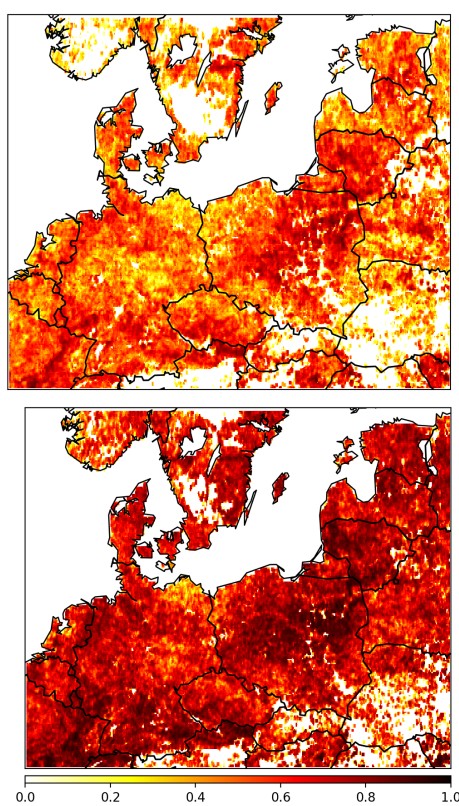

**Figure A4.** Performance of the temporal RF model in predicting $EVI_{anom}$, given by the out of bag scores. The top panel shows the scores for the climate-driven RF model and the bottom panel the corresponding results for the same model, but including $EVI_{anom}^{yr-1}$ as an additional predictor.



**Figure A5.** Importance of the four predictors used in the RF model to predict $EVI_{anom}$, spring (left) and summer (right), $SM_{anom}$ (top) and $T_{anom}$ (bottom), calculated from the Shapley additive explanation values (Methods).





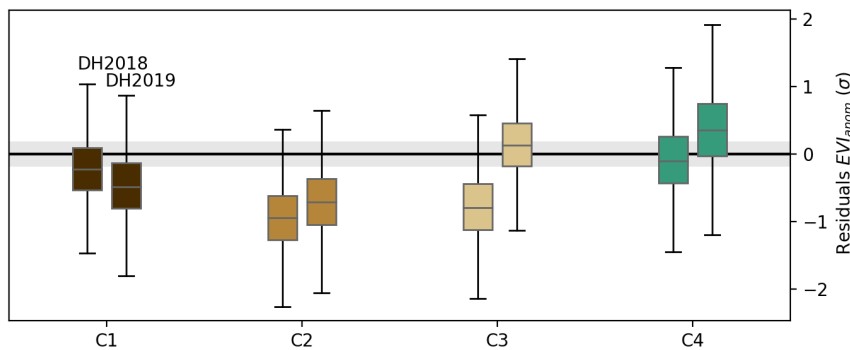

**Figure A6.** As in Fig. 5 bottom panel, but for the RF model trained using spring and summer $SM_{anom}$ and $T_{anom}$ as predictors, as well as $EVI_{anom}^{yr-1}$.

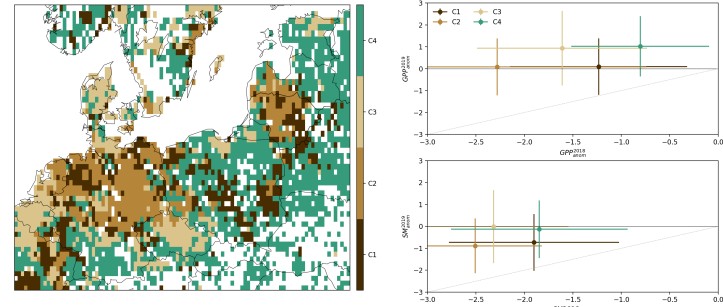

**Figure A7.** The left panel shows the spatial distribution of the four clusters from unsupervised classification of $(EVI_{anom}^{DH2018}, EVI_{anom}^{DH2019})$ values remapped to the coarser grid of LSMs. The corresponding $(GPP_{anom}^{DH2018}, GPP_{anom}^{DH2019})$ values simulated by the multi-model mean in each cluster are indicated in the top right panel (circles indicate the spatial mean and the lines spatial standard deviation within each cluster). The corresponding distribution of simulated $SM_{anom}$ pairs in each cluster are shown in the bottom right panel. The grey line, indicates similar anomalies in the two DH events.

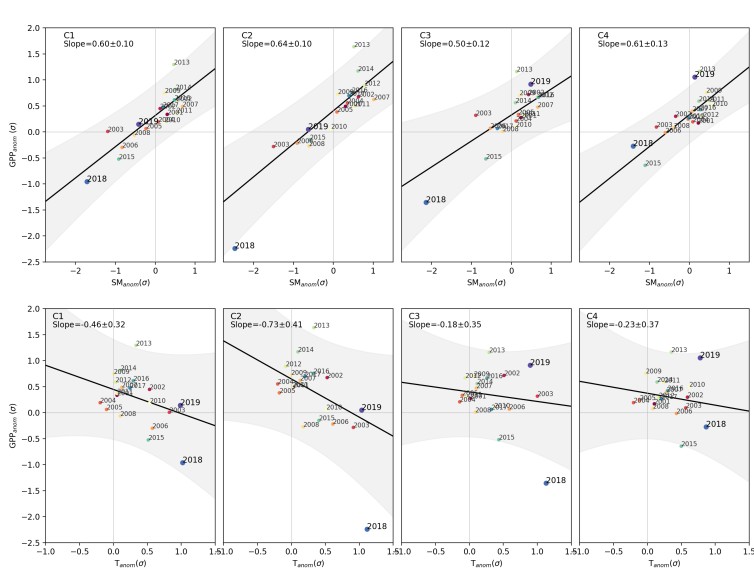

**Figure A8.** Same as Fig. 4 but for GPP and soil-moisture anomalies simulated by a subset of land-surface models from (Bastos et al., 2020a) extended up to 2019.