# Peer review of "Vulnerability of European ecosystems to two compound dry and hot summers in 2018 and 2019"

_Earth System Dynamics, 2021_

## Referee Comment (RC1)

**Review of "Increased vulnerability of European ecosystems to two compound dry and hot summers in 2018 and 2019" by Bastos et al.**

This work focuses on understanding the central European ecosystem response to two consecutive extreme dry and hot summers in 2018 and 2019. The authors combine different observation/reanalysis datasets and both statistical and dynamic vegetation modelling approaches to comprehensively evaluate the impacts of these compound events. Based on different responses to DH2018 and DH2019, 4 categories of ecosystems are identified, of which two show stronger negative effects of the recent DH events on vegetation compared to previous climate-vegetation linear relationships. One of the two classes shows increased ecosystem vulnerability to compound DH events possibly modulated by long-term warming and increased pCO2, and is characterized by grasslands and crops. The second class shows continuing browning from DH2018 to DH2019 even with alleviation of soil moisture deficits and heat stress in DH2019, mainly attributed to legacy effects of DH2018, and is characterized by forests and grasslands. These legacy effects, however, are missing in the current generation land-surface models, suggested by factorial simulations using an ensemble of these models. This may shed doubt on the ability of model skills to predict the resilience of temperature ecosystems to more likely compound events in the future.

The manuscript is clearly written and easy to understand its message, and the results will be of great interest to a wide community of climate and ecosystem researchers. However, I suggest a few points be addressed before it is considered for publication, as follows.

My biggest concern is that the "increased vulnerability" can be exaggerated. First, the authors only focused on the area with negative EVIanom in DH2018 (area_clusters), while the rest area (white color pixels in Fig. 3) was not considered, including core regions such as scattered parts in Germany and Poland, and large assembled parts in southern Sweden and the southeast of the study region. Based on the climate-vegetation relationship estimated from this area_clusters, the 2018 and/or 2019 EVIanom of some ecosystem clusters appears to be deviated from the long-term relationships. However, including the rest area can both affect the departures of 2018/2019 EVIanom as well as the long-term relationship. First, the slopes (EVIanom-SManom slope and EVIanom-Tanom slope) can be larger as modulated by vegetation susceptibility in the rest area during 2001-2017, for instance, the southeastern part (southern Sweden) during 2003 (2015) heat wave. Second, including 2018 EVIanom in the rest area can surely offset some of the 2018 departures since it is a positive EVI anomaly. Therefore, in my opinion the approach to just look at DH2018 EVIanom-negative region in this manuscript does not allow to conclude the increased vulnerability of European ecosystems to the compound events (also given the fact that it only covers 20% of the area_clusters).

Soil moisture anomaly is a useful indicator for the climate impacts of DH events. I feel the correlation is not enough to assess the model skills in simulating soil moisture variability (Table 1). First, the record can be too short for a robust correlation (Apr-Sep, sample size=6?), and this can be an issue not only for soil moisture validation, but also for GPP (e.g. L310). Second, even if the correlation is high, the magnitude of the changes in the absolute soil moisture values can be underestimated in the model due to too shallow soil. Therefore, calculating RMSE or normalized-RMSE using the absolute soil moisture values can be useful.

**Other comments**

L16 Should it be dominated by forests and grasslands? It is inconsistent with results shown in L240.

L179&L289 Please add a few lines to justify the use of the RF regression. What's the advantage compared to a linear multivariate regression?

L181 Please explain more on the 3x3 and 17x9 used here.

Sec 3.2.2 & Sec 3.2.3 There are some repetitions in these two sections, such as the definition of residuals.

L245 Is it a correct statement? It seems like a few exceptions can be there (SManom and Tanom of cluster C4).

L260-261 This may be inaccurate. See my major comment.

Fig. 4 Anomalies during EVIanom positive years and EVIanom negative years are supposed to be comparable (add up to 0 eventually). There is a trick that 2018&2019 were not included in this long-term relationship. If they were, "abnormal" positive values could show up.

L266 How can different out of bag scores affect explained variability so much?

L291-293 Please rephrase. The improved predictivity of RF seems to be contradictory to the comparable residuals.

Fig.6 The x-axis labels are hard to read.

Sec 5.2 Is the simulation of 2018 productivity anomalies really so well? At least during the spring season precondition it is not so consistent between model and data.

Fig. A5 I cannot find anywhere in the text that this figure is discussed.

**Technical comments**

L3 though -> through
L72 Modify the citation.
L180 Double-check the variables used here.
L226 excepting -> except
L285 Add . before "In DH2019"
L301 ,since -> , since
L310 GPP should be GPPanom?
L323 EVIanom, subscript anom
L388 related -> be related

---

## Author Comment (AC1)

**Increased vulnerability of European ecosystems to two compound dry and hot summers in 2018 and 2019**

Bastos, A., Orth, R., Reichstein, M., Ciais, P., Viovy, N., Zaehle, S., Anthoni, P., Arneth, A., Gentine, P., Joetzjer, E., Lienert, S., Loughran, T., McGuire, P. C., O, S., Pongratz, J., and Sitch, S. *Earth Syst. Dynam. Discuss*

**Response to Reviewer #1**

**R1C1: The manuscript is clearly written and easy to understand its message, and the results will be of great interest to a wide community of climate and ecosystem researchers. However, I suggest a few points be addressed before it is considered for publication, as follows.**

We thank the reviewer for the overall positive evaluation of our study and for the constructive comments. Below, we provide a point-by-point reply to the reviewer comments.

**R1C2: My biggest concern is that the "increased vulnerability" can be exaggerated. First, the authors only focused on the area with negative EVIanom in DH2018 (area_clusters), while the rest area (white color pixels in Fig. 3) was not considered, including core regions such as scattered parts in Germany and Poland, and large assembled parts in southern Sweden and the southeast of the study region. Based on the climate-vegetation relationship estimated from this area_clusters, the 2018 and/or 2019 EVIanom of some ecosystem clusters appears to be deviated from the long-term relationships. However, including the rest area can both affect the departures of 2018/2019 EVIanom as well as the long-term relationship. First, the slopes (EVIanom-SManom slope and EVIanom-Tanom slope) can be larger as modulated by vegetation susceptibility in the rest area during 2001-2017, for instance, the southeastern part (southern Sweden) during 2003 (2015) heat wave. Second, including 2018 EVIanom in the rest area can surely offset some of the 2018 departures since it is a positive EVI anomaly. Therefore, in my opinion the approach to just look at DH2018 EVIanom-negative region in this manuscript does not allow to conclude the increased vulnerability of European ecosystems to the compound events (also given the fact that it only covers 20% of the area_clusters).**

We agree with the reviewer that the expression "increased vulnerability" does not fully reflect the results of our study, which shows differentiated responses to DH2018 and DH2019. We propose dropping the expression "increased" from the title. We have added one sentence on regional compensation effects in the discussion as follows:

> *It should be noted, though, that we focused on pixels which were negatively impacted by DH18, but some pixels in the regional domain selected showed greening, even in DH18 (Fig. 1) These regional asymmetries result in partial regional compensation of the DH18 impacts, as shown in Bastos et al. (2020b).*

Nevertheless, we would like to justify our choice to focus on negative anomalies from DH2018.

Different perspectives can be used to analyse the impacts of a given extreme event, from a hydrometeorological perspective (Flach et al., 2018; Bastos et al., 2020a, b) to an impact-centered perspective (Smith, 2011; Bastos et al., 2014; Zscheischler et al., 2014; Reichstein et al., 2013). We agree with the reviewer that the choice of perspective matters, and we argue that this choice should be based on the specific question(s) one intends to answer.

First, we are interested in evaluating the impacts of the two consecutive extreme summers on vegetation activity, particularly how the impacts of the first event might precondition the response to the second event (Ruehr et al., 2019; Anderegg et al., 2020). Therefore, we follow an impact-centered compound event approach (Zscheischler et al., 2020), where we separate the impact of the concurrent hazard (high temperature and drought) from preconditioning or modulating factors. The preconditioning/modulating factors are expressed by departures of observed impacts from those predicted based on the hazard intensity only. This is an analogue approach to that used in (Bastos et al., 2020a), but their study was only modelbased. Here, using a data-driven approach, we find consistent results with their modelling study, particularly the higher vulnerability of cropland dominated regions to DH2018 (as well as DH2019).

Of the four clusters, only cluster 4 does not show departures from the long-term relationship, which can be associated with higher resistance and resilience of forests in response to DH2018 and DH2019. The other three clusters all show signs of increased vulnerability (L259-L262, Fig. 3), C1 to DH2019 only, C2 to both events, and C3 to DH2018 only. These three clusters encompass the majority of the studied pixels (56% of the area with negative EVI anomalies in DH2018), not 20%.

Second, indeed some regional compensation is expected, which would necessarily affect the relationships found *if* we would include the pixels with positive EVIanom in DH2018 in our regression analysis, which we do not. Moreover, regional asymmetries and compensation effects during DH2018 have been thoroughly analysed in (Bastos et al., 2020a, b).

Finally, we do not limit our analysis to the regional aggregated values, but also perform the same analysis at pixel level, so that regional compensation effects do not affect our results based on Figures 4 and 5. We noted however, that in Fig. 4 and 5, the pixels with positive anomalies in DH2018 were not masked out. This has now been corrected for consistency.

**R1C3: Soil moisture anomaly is a useful indicator for the climate impacts of DH events. I feel the correlation is not enough to assess the model skills in simulating soil moisture variability (Table 1). First, the record can be too short for a robust correlation (Apr-Sep, sample size=6?), and this can be an issue not only for soil moisture validation, but also for GPP (e.g. L310). Second, even if the correlation is high, the magnitude of the changes in the absolute soil moisture values can be underestimated in the model due to too shallow soil. Therefore, calculating RMSE or normalized-RMSE using the absolute soil moisture values can be useful.**

Here we were particularly interested in how the models simulate the dynamics during the DH events, not the model skill to simulate soil-moisture variability in general. We agree with the reviewer though, that this information is also important, to include the RMSE. We now compare the modeled variables with observed SM and EVI using both the correlation and RMSE for the two DH events (i.e. Table 1, updated).

**Other comments**
**L16 Should it be dominated by forests and grasslands? It is inconsistent with results shown in L240.**
Yes, thanks for pointing out, it's corrected now.

**L179&L289 Please add a few lines to justify the use of the RF regression. What's the advantage compared to a linear multivariate regression?**
In the original manuscript (L178-180) we explained that:

> *"Because impacts on EVI could result from non-linear interactions between soil-moisture and temperature or from legacy effects from spring (Bastos et al., 2020a; Lian et al., 2020), we extend this analysis by random-forest (RF) regression […]"*.

We have now reformulated the paragraph to improve clarity:

> *"However, departures from a linear model could also result from non-linear interactions between soil-moisture and temperature or from legacy effects from spring (Bastos et al., 2020; Lian et al., 2020). Therefore, we extend this analysis to the pixel scale and further include non-linear effects and interactions by fitting a random-forest (RF) model with SManom and Tanom in spring (MAM) and summer SManom and Tanom as predictors.*

**L181 Please explain more on the 3x3 and 17x9 used here.**
We have restructured the description of the method. The sentence now reads:

> *To reduce the risk of over-fitting due to the small sample size (17 years) and large number of predictors (4), we fit the RF model on 3 pixels × 3 pixels moving windows centered around each pixel (i.e. 17×9 samples).*

**Sec 3.2.2 & Sec 3.2.3 There are some repetitions in these two sections, such as the definition of residuals.**
We have removed redundancies in the revised version.

**L245 Is it a correct statement? It seems like a few exceptions can be there (SManom and Tanom of cluster C4).**
Thanks for pointing out the inconsistency. We were referring to the centroids of the clusters, but indeed some pixels show different behavior. We re-wrote the paragraph:

*All clusters align along proportional SManom and Tanom in DH2018 vs DH2019, with predominantly negative SManom and positive Tanom in both DH events but alleviation of soil-moisture deficits and heat stress in DH2019 compared to DH2018 (Fig. 2). […] in spite of drought and heat stress alleviation (Fig. 2). Furthermore, the distributions of climate anomalies for each cluster overlap each other and, in some cases, the 1:1 line, indicating that the intensity of the hazard (temperature, drought) cannot account for the resulting impacts alone.*

**L260-261 This may be inaccurate. See my major comment.**
We agree, the sentence was incorrectly formulated. We have now rephrased to:

*The results correspond to a general summer water-limited regime, especially in clusters $C_{Decline}$, $C_{HighV}$ and $C_{PRecov}$, which show stronger sensitivities to Tanom and SManom (slopes in Fig. 4) and higher variance explained by both models (R2 0.58–0.68 for SManom and 0.49–0.55 for Tanom). For these clusters, EVIanom is below the 95% confidence interval of the long-term linear relationships for DH18 ($C_{PRecov}$ and $C_{HighV}$) and DH19 ($C_{Decline}$ and $C_{HighV}$). SManom and Tanom in DH18 and DH19 are generally similar to those of 2003, but DH18 was drier than 2003 in $C_{PRecov}$ and $C_{HighV}$.*

The justification to focus on these three clusters can be found in the reply to R1C2 above.

**Fig. 4 Anomalies during EVIanom positive years and EVIanom negative years are supposed to be comparable (add up to 0 eventually). There is a trick that 2018&2019 were not included in this long-term relationship. If they were, "abnormal" positive values could show up.**
We disagree with the reviewer that this is a "trick". The rationale for not including these years in the modelling approach is to evaluate whether the vegetation response to the hazard in each of these two years are consistent with long-term relationships with climate (2001-2017).
I.e., we are testing whether past climate-vegetation relationships can be used to predict the anomalies in each event. Such an approach has been used in other studies to evaluate legacy effects from droughts (Anderegg et al., 2015), as well as extreme vegetation anomalies (Bastos et al., 2017).

**L266 How can different out of bag scores affect explained variability so much?**
This refers not to different ways of calculating OOB, but to the differences in OOB between pixels as shown in Fig. A4. We have rephrased for clarity:

*"The model is able to explain 48-90% (median and maximum out of bag score across pixels) of the pixel-level temporal variability of summer EVIanom in 2001−2017 […]"*

**L291-293 Please rephrase. The improved predictivity of RF seems to be contradictory to the comparable residuals.**
It is not contradictory since it refers to the long-term RF model out-of-bag scores for the period 2001-2017, versus the ability to predict DH2018 and DH2019. See our reply to the comment on Fig. A4 page above.

**Fig.6 The x-axis labels are hard to read.**
We apologize for the low resolution of the figure. We have improved the readability of all figures.

**Sec 5.2 Is the simulation of 2018 productivity anomalies really so well? At least during the spring season precondition it is not so consistent between model and data.**
Thank you for pointing out. Indeed, if we include the full year the correlations are lower. However, we are also interested in how models simulate the impacts of DH2018 and DH2019, therefore we keep the analysis for the 6 months as well. We now include a new table where we compare the correlations and RMSE (R1C3) for the whole year.

**Fig. A5 I cannot find anywhere in the text that this figure is discussed.**
It is now refered to when analysing the RF model fit (Section 4.2).

**L3 though -> through**
Corrected.

**L72 Modify the citation.**
MULNV-NRW refers to Ministerium für Umwelt, Landwirtschaft, Natur- und Verbraucherschutz des Landes Nordrhein-Westfalen. Since it is correctly linked in the reference, we decide to keep it as in the original manuscript, except the reviewer has a specific suggestion.

**L180 Double-check the variables used here.**
We have rephrased for clarity:

> "[…] random-forest (RF) model with $SM_{anom}$ and $T_{anom}$ in spring (MAM) and in summer (JJA) as predictors (i.e. four predictors, $T^{spr}_{anom}$, $SM^{spr}_{anom}$, $T^{sm}_{anom}$, $SM^{sm}_{anom}$)."

**L226 excepting -> except**
Done

**L285 Add . before "In DH2019"**
Done

**L301 ,since -> , since**
Done

**L310 GPP should be GPPanom?**
Yes, it's corrected.

**L323 EVIanom, subscript anom**
Corrected

**L388 related -> be related**
Corrected

---

## Author Comment (AC2)

**Increased vulnerability of European ecosystems to two compound dry and hot summers in 2018 and 2019**

Bastos, A., Orth, R., Reichstein, M., Ciais, P., Viovy, N., Zaehle, S., Anthoni, P., Arneth, A., Gentine, P., Joetzjer, E., Lienert, S., Loughran, T., McGuire, P. C., O, S., Pongratz, J., and Sitch, S. *Earth Syst. Dynam. Discuss*

**Response to Reviewer #2**

**R2C1: The manuscript by Bastos et al explores the existence and extent of compound effects of subsequent dry and hot summers on vegetation, using the case of 2018 and 2019 in Europe. The manuscript is of general interest to a large audience, across several disciplines. Overall, the manuscript is generally well-written and clear (but see below). I have however three main suggestions**
We thank the reviewer for the evaluation of our manuscript and for their comments and suggestions.

**R2C2: The manuscript is dense, both in terms of methods and results. The results and even more so discussion are thus inevitably complex, but, unfortunately, at times not very clear, with apparently contrasting statements. A general suggestion would be to give descriptive names to the four clusters, to facilitate the reader throughout the text. Further, I list here some examples of apparently contrasting statements or unclear sentences, but I suggest that the entire discussion is revised for better clarity.**
We agree that the manuscript is dense. We thank the reviewer for pointing out areas for improvement and for the suggestions. We have now changed the names of the clusters to: $C_{Decline}$, $C_{HighV}$ (from high-vulnerability), $C_{PRecov}$ (partial recovery), $C_{Greening}$. We also revised the discussion in an attempt to improve clarity, and corrected the imprecisions pointed out by the reviewer below.

**R2C3: I find of particular relevance the question of the responses of different land uses. This point is relatively prominent in the abstract, but then only briefly touched upon in the results and discussion. I think the manuscript would gain in terms of impact (and interest within a broader audience) should this question be explored in more detail. To this aim, I would suggest redoing the analyses at the basis of Figure 6, but separating land uses. This could give some insights in how the different vegetation types respond to temporal compound events. Of particular relevance would be the correlation with previous year's EVI anomalies, temperature, soil moisture and soil available water capacity. This additional analysis would for example reduce the speculation behind statements like that in L349, by allowing exploring the effects of available soil water.**
We agree with the reviewer that evaluating the results by different land-cover/uses increases the relevance of the study. Separating between land-uses at the spatial scale of the EVI (5km) is challenging, though, because of the substantial, small scale landscape heteorogeneity in Central Europe. This is also the reason for our analysis of LC selectivity for each cluster, rather than simple categorization into LC types. To address the reviewers concern, we propose a new version of Figure 6, in which we show results separately for pixels with high tree cover vs. pixels with low tree cover (reproduced below). We also compare residuals for pixels with high and low tree cover fractions.
The updated figure supports a contribution of the warm spring legacy effects in explaining residuals in 2018, and a strong difference in the water-limitation (given by the relationship with SManom in summer) between pixels with high vs. low forest cover in 2018. Previous year's EVI becomes more important in 2019, as do the summer climate variables and isohydricity. We have accordingly revised all the results and discussion referring to Fig. 6.

[Figure]

Figure R1: Spatial partial correlation (spearman) between EVIanom residuals and environmental variables in DH18 (top panels) and DH19 (bottom panels), for pixels with high (dark green) and low (light green) tree cover. The variables considered are: spring and summer Tanom and SManom (indicated by superscripts *spr* and *sm*, respectively), EVIanom in the previous growing season ($EVI^{yr-1}$), plant isohydricity (IsoH) and the number of dry months (DM). Because of the large number of pixels considered, all correlations are significant (p-val<<0.01).

**R2C4: One could expect that crops, by being mostly annual plants, would have a radically different response to temporal compound events from all other perennial vegetation. Specifically, one could expect no substantial legacy effects, or even a positive EVI anomaly in 2019, as the result of reduced nutrient use and losses via leaching in the previous year, characterized by low production and low soil moistures. While the discussion is definitely more focused on vegetation dominated by perennial plants, there are few hints at crops also having some legacy effects. I think it would be helpful to discuss in more detail why this is the case (if this is the case), thus better grounding the results in our ecophysiological understanding of plant and ecosystem response to (repeated) heat and drought. The additional analysis suggested above could shed some clarity on this matter, making the discussion less speculative. Authors could also consider how the LSM performance in 2019 is affected by land use: it could be expected that, if LSMs fail to represent the carry over effects, then they would be performing better in ecosystems where carry over effects are intrinsically more limited.**

We agree with the reviewer that crops should not show strong legacy effects, at lest from a physiological perspective, and it was not our intention to convey this message. We have carefully revised the text to avoid such confusion. We find a high correlation with previous seaason's EVI for pixels with low tree cover, though, which is surprising. We added the following explanation in the revised manuscript:

> *The stronger correlation found in low tree cover pixels is surprising, as crops and many grasslands are mostly annual plants. The high correlation between EVIanom residuals and $EVI^{yr-1anomin}$ DH19*

*can indicate either that pixels strongly impacted by DH18 were associated with amplified impacts by DH19 (negative residuals), or that pixels affected moderately by DH18 (less negative $EVI^{DH18}$anom were associated with positive residuals, i.e. stronger recovery.*

*Damage to roots and tissues or depletion of carbon reserves from DH18 leading to higher vulnerability to DH19 could explain the positive correlation in high tree cover pixels in $C_{Decline}$. Conversely, the moderate DH18 impacts in $C_{Greening}$ may have resulted in increased resistance to DH19. The strong correlation found in low tree cover pixels is, though, surprising, as European crop species tend to be annual plants, and annual species can also be found in many grasslands. For these pixels, it is more likely that the positive correlation is explained by management practices, e.g. through earlier harvest or active reduction of stand density in DH19 (Bodner et al., 2015).*

Evaluating the performance of the LSMs per land-cover type would require being able to separate the land-covers in the observations. In the study region, the maximum grid-cell average forest cover is 70% and only 4% of the 5x5 km² pixels have forest cover above 60%. Therefore, when remapping to the 25km resolution of the LSM models, one would be left with too few pixels, if any, for appropriate model evaluation.

**L313: Isn't the high correlation in contrast with the statement in L 305?**
Indeed, there is an apparent contradiction, but these two sentences refer to different patterns: average anomalies in the two DH events (L305) and the temporal evolution of the anomalies (L313). In line with R1C3, we now include RMSE as well. We agree, however, that these points were not clear. In the restructuring of the results and discussion, we moved this point to the discussion section and rephrased as:

*The LSMs perform well in simulating the magnitude and evolution of productivity anomalies in 2018, but not in 2019. The recovery simulated by LSMs in DH19 can be partly explained by a strong recovery of modelled soil-moisture (Fig. B7), but may also result from limited ability of LSMs in simulating changes in ecosystem vulnerability during the two DH events. The latter is supported by the fact that simulated SManom shows good agreement in the temporal evolution of soil-moisture anomalies with both observation-based datasets but not of GPPanom (Table 1).*

**L322-324: This is an important point, but it is not at all clear in the text.**
We have now emphasized this point in the revised version of the discussion:

*In DH18, we find a positive effect of spring warming in vegetation growth, leading to weaker departures from long-term vegetation--climate relationships (observed EVIanom more positive or less negative than modelled), but with associated water depletion amplifying the impacts of DH18 in summer in pixels with low tree cover. These results are in line with Bastos et al. (2020a) that showed contrasting seasonal legacy effects of warm springs in crop versus forest dominated regions.*

*On the contrary, spring and summer $T^{sm}$anom in 2019 (or cooling, see Fig. B1) are negative correlated with EVIanom residuals in both high and low tree cover pixels. This indicates increasing damage from heat stress, for example due to reductions in evapotranspirative cooling (Obermeier et al., 2018) or cascading impacts of compound heat and drought, such as insect attacks (Rouault et al., 2006).*

**L349: I think this is a potentially controversial point. Deeper roots are an advantage, if the off season has provided water recharge.**
In the figure below, we show the mean value of total water storage over the study region from the GRACE Data Analysis Tool. The vertical line indicates January 2019, where it can be seen that water equivalent values had returned to values registered only before DH2018.

[Figure]

Water Equivalent Thickness – Land (GRACE, GRACE–FO JPL)
Source: GRACE, GRACE–FO
47.0000N, 3.0000E – 60.0000N, 29.0000E
Feb 2010 – Dec 2019

● Water Equivalent Thickness – Land (GRACE, GRACE–FO JPL)

**L354-355: On which basis can it be stated that this is 'consistent'?**
We meant that forests and grasslands tend to have higher isohydricity than croplands (Konings and Gentine, 2017). The discussion section has been restructured (R2C2) and the sentence in the meantime removed.

**L356: Why is the vulnerability increased? Is this because of heat? Isn't this in contrast to L350? Does this apply just to 2019? And, even if so, how could an opposite response in the two years be justified?**
Indeed, the formulation was confusing. The results for DH2019 are opposite than for DH2018, which is a key result of this analysis. We have reformulated the discussion and hope the message is now clearer:

> *In DH18, we find a positive effect of spring warming in vegetation growth, leading to weaker departures from long-term vegetation--climate relationships (observed EVIanom more positive or less negative than modelled), but with associated water depletion amplifying the impacts of DH18 in summer in pixels with low tree cover. These results are in line with Bastos et al. (2020a) that showed contrasting seasonal legacy effects of warm springs in crop versus forest dominated regions.*
> *On the contrary, spring and summer $T^{sm}$anom in 2019 (or cooling, see Fig. B1) are negative correlated with EVIanom residuals in both high and low tree cover pixels. This indicates increasing damage from heat stress, for example due to reductions in evapotranspirative cooling (Obermeier et al., 2018) or cascading impacts of compound heat and drought, such as insect attacks (Rouault et al., 2006).*

**Minor comments:**
**There are several imprecisions in the text (missing or misplaced blank spaces, inconsistencies in symbols, etc.), in particular in the results and discussion sections.**
We have corrected the typos.

**The readability of Fig 3 would be greatly improved if it was bigger (in particular the right panels). Also, could a different set of colors be used, to highlight differences in the map? C2 and C3 are difficult to distinguish now.**
We have now increased the size of the figure and adapted the color scheme so that C2 and C3 can be more easily distinguished (C2 is now redder). We have also checked that the color scheme is colorblind friendly.

**Also Fig. 4 could be a bit bigger, possibly with larger and differently shaped symbols for 20018 and 2019 (which are anyhow outside the regression).**
We now use bigger and different shaped symbols for 2018 and 2019.

**L131: Why not also June-August 2019?**
The goal of these simulations was to test the hypothesis that models might simulate too weak impacts of DH2019 because they lack legacy effects from summer 2018. The added value of running one additional simulation with fixed summer 2019 climate is, in our opinion, limited, since we expect concurrent responses to be similar, in the absence of legacy effects. Since such simulations are time consuming, we propose not to include such an experiment.

**L172: In which sense there is an acclimation to drought, at the scale of one-to-two years?**
Certain responses to DH18 could confer greater resistance to DH19. We have added now:

> *Impaired functioning during the recovery period can additionally increase the hazard of subsequent disturbances, e.g. insect outbreaks (Rouault et al., 200^). However, reductions in leaf area, increases in root allocation (McDowell et al. 2008) or reduced growth, by reducing evaporative tissue and enhancig water uptake capacity, could also confer an advantage to subsequent droughts (Gessler et al., 2020).*

**L244: I think the term 'recovery' is not necessarily correct, at least not in all ecosystems. So, I suggest using a more neutral 'less negative EVI anomalies'.**
We refer to "partial recovery", not full recovery, which is justified by all pixels in C3 showing higher (less negative) EVI anomalies in 2019. For readability, we suggest keeping as is.

**L 285: 'stronger' with respect to what? DH2018?**
Yes we meant with respect to DH2018. This is now corrected.

**L357: I find the term 'natural ecosystems' potentially confusing here. I suppose it is used to contrast forests/grasslands to croplands, but, in Europe, very few forests can be considered natural (in the sense of unmanaged) ecosystems. Their response to heat and drought is certainly mediated by species choice and other aspects of management.**
Thanks for pointing out, we agree. This sentence has been removed in the revised version of the manuscript.

---

## Author Comment (AC3)

**Increased vulnerability of European ecosystems to two compound dry and hot summers in 2018 and 2019**

Bastos, A., Orth, R., Reichstein, M., Ciais, P., Viovy, N., Zaehle, S., Anthoni, P., Arneth, A., Gentine, P., Joetzjer, E., Lienert, S., Loughran, T., McGuire, P. C., O, S., Pongratz, J., and Sitch, S. *Earth Syst. Dynam. Discuss*

**Response to Reviewer #3**

**The manuscript talks about a relevant topic showing the importance of consecutive dry and hot events for ecosystems. The paper investigates why temporally-compound compound extremes can amplify the damage to the ecosystem focusing on 2018 and 2019 in Europe.The paper addresses a relevant scientific question and it is within the scope of ESD with the potential to have an audience from a broader community of climate impact studies, agricultural systems, hydrology, and multi-sector dynamics.The paper concludes that process-based models miss the legacy effects of consecutive compound hot and dry summers. The final results are based on a) regression and correlation analysis, and b) LSM outputs. However, the data preparation has heavily relied on satellite scans from MODIS and simulation outputs of Land Surface Models.**

We thank the referee for the constructive review of our study.

**R3C1: The paper describes the data sources in great detail which is good. I understand that this is a critical part of the research, but it distracts the reader from the main message. I suggest moving the data construction information to the appendix.**

We will move the description of the simulation protocol to the Annex and we have attempted to condense the description of the observation-based datasets, since they are published elsewhere.

**R3C2: Finally, I suspect the results are reproducible. It would be beneficial if the authors could share the major constructed datasets for regression analysis and related outputs of LSM models.**

All datasets used in the paper are freely available, at exception of the extension of the LSM outputs for 2019. The SoMo.ml dataset has been published recently (https://www.nature.com/articles/s41597-021-00964-1). We have added a section on data availability with the links to the public repositories and will make the model outputs for 2019 available as supplement to the paper.

**R3C3: The title of the paper talks about compound dry and hot summers. The model and data do not include metrics of compound extremes. The T and SM are often separated in the paper and are considered individually. I expected to see some compound indicators. I am not sure the term "compound" in the title is well-represented within the paper.**

The whole analysis proposed here is based on the conceptual framework of compound event analysis. From a meteorological perspective, DH2018 and DH2019 extreme summers were "compound events" in that both high temperatures and strong drought conditions were observed. Each taken individually can be considered a multivariate compound event (Zscheischler et al., 2020). Additional effects that make each of these events "compound" from an ecological perspective are the preconditioning effects of the warm/sunny spring in 2018 (Bastos et al., 2020a), and of the impact of DH2018 in preconditioning the response to DH2019.

An expanded version of our Figure 1 is shown below:

[Figure]

Where green represents drivers of hazards, blue the hazards, red the impacts, and yellow preconditioning effects (Zscheischler et al., 2020).

In the paper, T and SM are considered separately only in the specific case of the linear regression shown in Figure 4. For the subsequent analyses, T and SM are analyzed jointly as predictors of EVI anomalies. The goal of the analysis in Fig. 4-6 is to separate the preconditioning effects of DH2018 in explaining the EVI anomalies in DH2019.

We acknowledge that the text might not have been clear in this respect, and we made an effort to improve the clarity in the revised version of the manuscript:

> *From a hydrometeorological perspective, the dry and hot summers in 2018 and 2019 (DH18 and DH19, respectively) could be considered individually as two compound events in that both high temperatures and strong drought conditions were observed (Zscheischler and Fischer, 2020) Taken together, they could also be analysed as a temporally compound event (Zscheischler et al., 2020).*

**R3C4: The paper is written well. Still, the overall presentation requires revisions. Please describe the main variables in more detail in a table in the appendix. The reader deserves a clear description of the main variables of this study and the main model evaluating the relationship between them (e.g. variables explaining EVI in regression models). There is no single equation neither a table showing the underlying model of the study. While the main work is based on regressions and correlation analysis, the reader should wait until page 7 to learn about them. The problem is that the regression strategy has a vital role here. Are you estimating the marginal impacts for each pixel or a set of clusters? These should be clarified with a written equation with clear indexing of all the variables. In addition, this can be a useful reduced-form model for future studies. In its current form, the regression section looks pretty weak.**

We have now re-written the methods section explaining the regression analysis. We would like to note that the methodology was also mentioned in the abstract. We have nevertheless revised the abstract and introduction to clarify the approach used.

**R3C5: There is no discussion on the goodness of fit for regression and the causal analysis. The lagged EVI used in the model, while can be used to prove the existence of the legacy effect, does not tell us the exact sources of legacy effects. There is a high chance of omitted variables here (e.g. soil moisture in lower layers or disease as discussed in the paper). This could be briefly addressed in the appendix.**

Thanks for pointing out. Indeed, one of the drawbacks of the current approach is that is cannot fully separate between the sources of legacy effects. Arguably, some of these causes could be pinned-down if, for example, long-term spatially explicit data on disturbance sources, and specifically for 2019, would be available. We added a note on this in the revised version of the discussion:

> *Increased vulnerability may be explained by modulating effects of global change on vegetation condition (e.g., ``hotter droughts'' (Allen et al. 2015), Fig. 1) and, in the case of DH19, it may be further linked to inter-annual legacies from the impact of DH18. The first should be expressed by relationships between EVIanom residuals and climatic variables. The latter are more difficult to assess without comprehensive data about different competing factors, e.g. defoliation or damage from Ruehr et al. (2019), higher susceptibility to diseases and pests due to reduced health (McDowell et al. 2020) or increased hazard of insect disturbances due to warm conditions (Rouault et al., 2006). The relationships between EVIanom residuals and $EVI^{yr-1}$anom provide an approximation, but do not allow to identify the underlying drivers.*

Unfortunately, spatially explicit data about other variables is not available, especially not up to 2019. Additionally, other effects, such as physiological mechanisms explaining legacy effects may be very difficult to attribute at the large scale at which this analysis is performed (5-25km). Nevertheless, we hope that this approach can be adapted for local scale studies, in which some of the additional variables needed (deeper-level soil-moisture, sap-flow, vegetation structure and allocation to leaves, stems, roots, etc, information about pests and diseases) might be available.

**R3C6: Unfortunately, the interpretations and conclusions are more than what the model results show. For example, the claim in L316-319 is too strong. The correct conclusion is that "the proposed model" and current LSMs did not capture this legacy effect. This effect can be captured in future studies in models with different variables, metrics, and methods. This can be a shortage of the methods of this study. The results can be biased by omitting some variables (e.g. disease, nutrient, radiation, etc). This is the nature of science. Further investigation is required to have such a strong conclusion.**

With this study, we show that considering inter-annual legacy effects of a given extreme (DH2018) is important to understand the dynamics of vegetation in response to a subsequent extreme event (DH2019). Such effects have been conceptually described, e.g. in (Ruehr et al., 2019; Gessler et al., 2020), but quantifying them at large scales remains a challenge (Kannenberg et al., 2020). Our modelling approach based on EVI is designed to detect legacy effects, if existing, and not necessarily to model them. Our residual analysis (Fig. 6) provides some hints of possible variables needed to understand these effects, but is by no means an exhaustive list. More data is needed to test different hypotheses about sources of legacy effects (see comment above). On the other hand, such processes can be implemented in LSMs – some of them are under ongoing development – so that their relevance to the observed dynamics can be evaluated. In line with the comments from R2, we have thoroughly revised the results and discussion and we hope that this point is now clearer.

**R3C7: The paper employs the soil moisture data based on volumetric soil moisture of the top 28 cm. I expected to see other metrics of root-zone soil moisture depending on the vegetation dominance. The soil moisture in lower layers can be "a" major source of legacy effect which is ignored apparently. As mentioned in the paper: "total water storage was lower in 2019 due to the water storage deficit resulting from the 2018 event". Probably, a more precise hydrological product should be used to capture this.**

The soil-moisture dataset (SoMo.ml) used here represents the top 50cm, and thereby already a significant portion of the root-zone. Furthermore, it is to be expected that the temporal variability in (slightly) deeper layers is similar and thereby somewhat reflected in our analysis. While we agree with the reviewer that it would be preferable to use an observation-based soil moisture product covering even deeper layers, we are not aware of such a dataset. For example, total water storage from GRACE includes groundwater, snow, lakes and rivers, so not really the plant-available water. Other remote-sensing based datasets (e.g. ESA-

CCI, SMOS) are limited to the surface layer. Therefore, we think that SoMo.ml is, in our perspective, the most adequate product to use here.

**Other issues.**
**Define vulnerability**
Vulnerability is now defined:

> *Vulnerability to DH is defined as the impact of the physical hazard (hot and dry conditions) on vegetation and assessed by remotely-sensed EVI and modelled GPP anomalies.*

**L3: "though" does not seem the right word here.**
Corrected.

**L10: Please revise the sentence.**
It now reads: *"These estimates correspond to expected EVI anomalies in DH18 and DH19 based on past sensitivity to climate."*

**L29: "hot and dry" is better to be replaced by "dry and hot" to better represent the DH abbreviation.**
Done.

**L65: extra parentheses**
Corrected.

**L124: This is the data section. Maybe change the title to show this.**
Corrected.

**L268: is this your finding? If not move it to the discussion.**
We rephrased for clarity: *"[…] and estimates summer water limitation and negative legacy effects from spring warming, consistent with process-based modelling studies."*

---

## Author Response (AR2)

**Vulnerability of European ecosystems to two compound dry and hot summers in 2018 and 2019**

Bastos, A., Orth, R., Reichstein, M., Ciais, P., Viovy, N., Zaehle, S., Anthoni, P., Arneth, A., Gentine, P., Joetzjer, E., Lienert, S., Loughran, T., McGuire, P. C., O, S., Pongratz, J., and Sitch, S. *Earth Syst. Dynam. Discuss*

**Response to reviewers**

**Reviewer #1**

**R1C1: I appreciate the authors' efforts in responding to all of my initial comments and suggested edits. I recommend the revised manuscript for publication in Earth System Dynamics.**

Thank you for acknowledging our effort and the constructive review, which helped improving the manuscript substantially.

**Reviewer #2**

**R2C1: The streamlined version of the methods and the revised presentation of results are now clearer and easier to follow. I have just a handful of remaining comments, which I list below.**

We appreciate the reviewer's positive evaluation of our revision effort. We address the additional comments below.

**R2C2: "Compound event" is (implicitly) used with two meanings in the manuscript. One is relative to co-occurring dry and hot conditions. The other refers to the occurrence of these conditions in two subsequent here. I suggest making these two meanings explicit, before specifying that the focus is on the temporal compounding of compound events. This is because co-occurring dry and hot conditions could also be of interest, given their multiplicative effects on vegetation (the 'hotter drought' now mentioned more prominently; see e.g. Suzuki et al 2014 New Phytologist), but not the focus of this work. I think this would be of help in underlining where the main novelty lies.**

Thank you for pointing out that this was not clear, and for the excellent reference. We have revised Figure 1 (new version reproduced below) as well as the last paragraphs of the introduction, where these concepts are introduced:

> "From a hydrometeorological perspective, each of the dry and hot summers in 2018 and 2019 (DH18 and DH19, respectively) can be considered a multivariate compound event in that both high temperatures and strong drought conditions were observed (Zscheischler and Fischer 2020). Taken together, they can be considered a temporally compound event (Zscheischler et al. 2020). For example, Boergens et al. (2020) have shown that while soil-moisture deficits in summer 2019 were not as pronounced as in 2018, total water storage was lower in 2019 due to the water storage deficit resulting from the 2018 event. Given the unprecedented magnitude of DH18, it is likely that at least some ecosystems had not yet fully recovered in 2019. Therefore, from an ecological perspective, DH19 could additionally be considered a preconditioned compound event, where the impact of DH18 may affect the response to DH19 (Fig. 1)."

> "Using both remote-sensing data and an update of the simulations by (Bastos et al., 2020a), we attempt to separate these different effects, namely: how the combination of hot and dry conditions affected the vulnerability of ecosystems to the two events (multivariate compound event), how the repetition of a dry and hot summer affected the response to DH19 (temporally compound event)

*and how inter-annual legacy effects due to impacts of DH18 affected ecosystem vulnerability to DH19 (preconditioned compound event).*

*We first use a statistical modelling approach to evaluate whether signs of increased vegetation vulnerability to DH18 and DH19 can be found and to attribute changes in vulnerability to inter-annual legacies and other modulating effects. We then compare observation-based results to updated simulations by state-of-the-art land-surface models and dynamic global vegetation models (for simplicity referred to as LSMs) designed to isolate the impacts of DH18 and legacy effects (Bastos et al., 2020a).*"

[Figure]

Figure R1: New version of Figure 1. For simplicity, we removed the global change box, and make the two multivariate compound events (DH18 and DH19) more explicit. We also follow a color code more similar to Zscheischler et al. (2020) to facilitate the interpretation of the figure.

**R2C3: In Fig. 2 it would be helpful to specify the meaning of the rectangle. Is that the study region? Or was that defined based on the occurrence of the conditions, as written in the caption?**
The rectangle corresponds to the study region, which is the domain affected by two consecutive hot and dry summers. This is now clarified in the figure caption.

**R2C4: In the definition of tree cover (Fig. 6), does the low tree cover refer to the 5% lower or lowest tree cover, i.e., the 5% of pixels that have the lowest tree cover in the dataset? How much does this translate to, in terms of tree cover, approximately? And how about the top 5%? I think it is important to give a sense of what tree cover level the bars refer to.**
Thanks for the important suggestion. We added the following sentence in Fig. 6 caption:
    *"High TC pixels have tree cover fractions above 58% and low TC have virtually no trees (TC<0.4)."*
We also added small precisions to the text when referring to low TC pixels.

**L374 and elsewhere: 'tree cover' as opposed to 'forest cover' for clarity and consistency**
Thanks for noting the inconsistency, it is now corrected.

**Reviewer #3**

**R3C1: The paper reads much better now. The results and discussion sections are improved significantly.**
**My only minor comment is to spell out "JJA" and "MAM" for the general audience.**
Thanks for the feedback. We now spell out the months when JJA and MAM are introduced.

**R3C2: Also, I would like to see one or two paragraphs be added as the implications for interdisciplinary research and future research.**

We have accordingly rephrased the conclusions section:

*The summers of 2018 and 2019 were both exceptionally hot and dry over Central Europe, and both were associated with widespread vegetation browning and tree mortality events. Here we propose an approach that analyses this event as a combination of three types of compound events Zscheischler et al. (2020) that consider: (i) the compound effects of hot and dry conditions; (ii) the effect of repeated stress conditions in 2019 and (3) the legacy effects from DH18 impacts in preconditioning the impacts of DH19. Using statistical and process-based modelling, we quantify these effects and identify modulating effects, e.g. by land-cover composition. This approach can be extended to other types of events that may not fall in a single type of compound event.*

*Based on remote-sensing data, we find signs of degradation trajectories in 20% of the study area, with vegetation browning in spite of drought alleviation in DH19. We showed that inter-annual legacies from DH18 played an important preconditioning role in amplifying the impacts of DH19. While LSMs simulated well the impacts of the first event (DH18), they showed limited skill in simulating the impacts of the subsequent compound event (DH19).*

*Our results show that compounding effects of multiple and repeated stressors and ecological dynamics can result in non-linear and unexpected impacts Schuldt et al (2020), that LSMs still cannot realistically simulate. Attribution of inter-annual legacy effects from DH18 and of LSM errors to internal processes (e.g. drought-induced damage and mortality) or others such as insect outbreaks remains challenging because of up-to-date datasets on tree mortality, tree carbon reserves or spatially-explicit information on biotic disturbances are very limited.*

*Since extreme DH events are projected to become more common in the coming decades, better understanding the interactions and feedbacks between climate extremes, natural disturbances and ecosystem dynamics is fundamental to anticipate threats to the stability of forests in the temperate regions and elsewhere. Overlooking these effects may result in an overestimation of the resilience of the $CO_2$ sink to climate change.*